# Identifying and Addressing Delusions for Target-Directed Decision-Making

## Abstract

Target-directed agents utilize self-generated targets, to guide their behaviors for better generalization. These agents are prone to blindly chasing problematic targets, resulting in worse generalization and safety catastrophes. We show that these behaviors can be results of delusions, stemming from improper designs around training: the agent may naturally come to hold false beliefs about certain targets. We identify delusions via intuitive examples in controlled environments, and investigate their causes and mitigations. With the insights, we demonstrate how we can make agents address delusions preemptively and autonomously. We validate empirically the effectiveness of the proposed strategies in correcting delusional behaviors and improving out-of-distribution generalization.

## 1 Introduction

An important use of Reinforcement Learning (RL) is to learn skills generalizable for application, after training from limited training tasks. Despite numerous efforts to bridge the "generalization gap" between performance during training and evaluation, generalization abilities of RL agents remain largely unsatisfactory. Recent works attributed this to existing agents' lack of *reasoning* abilities to face Out-Of-Distribution (OOD) changes (Di Langosco et al., 2022).

Thus, embracing the concept of intentions, a certain type of decision-time planning agents are developed, which make use of their more adaptive planning outcomes, which we call *targets*, to direct its behaviors to generalize better in novel situations (Alver & Precup, 2022). For instance, these targets can help decompose a complicated task into small and familiar steps. In this paper, we use *targets* to denote (sub-)goals produced by the agents themselves during decision-time planning. These "target-directed" agents make use of *generators* to propose candidate targets, as well as optional *estimators* to evaluate the favorability of the candidates, to select a target to commit to.

While target-directed agents were supposed to be able to generalize well OOD, they are often observed to be blindly chasing unreachable or unsafe targets (Bengio et al., 2024). This paper focuses on the investigations of these behaviors, which existing literature has largely neglected. We introduce the readers to a new perspective to show that these agents are likely designed to be delusional. Unlike hallucinations, *delusions* are obviously wrong beliefs that an agent holds, and reflect an *inability to reject false beliefs* (Kiran & Chaudhury, 2009). These in the RL context correspond to false beliefs that are *natural* results of an agent's improper learning process, which limit the agents' situational understanding and lead to worse generalization or even safety catastrophes. Delusional behaviors stem from the incoordination of the *generator* and the *estimator*: problematic targets should not be generated, and even if they were, they should not be favored (Corlett, 2019).

This paper discusses some necessary conditions for agents to address delusions autonomously and preemptively: 1) incorporation of an estimator, enabling the rejection of problematic targets; 2) adequate learning rules with which false beliefs about targets can be counteracted and 3) proper training data that can counteract delusions. With clear diagnoses in the controlled environments, we identify types of delusions, and investigate proper designs of the training processes. We then applied our ideas to hindsight relabeling, a fundamental form of target-directed RL training, and validated experimentally that our strategies led to lower delusion-related errors and reduction in delusional behaviors, resulting in significant improvements in OOD generalization performance.

## 2 PRELIMINARIES

**RL & Problem Setting.**: An RL agent takes actions to interact with an environment to gain rewards. The interaction may be modeled as a Markov Decision Process (MDP) $\mathcal{M} \equiv \langle \mathcal{S}, \mathcal{A}, P, R, d, \gamma \rangle$, where $\mathcal{S}$ and $\mathcal{A}$ are the sets of states and actions, $P : \mathcal{S} \times \mathcal{A} \rightarrow \text{Dist}(\mathcal{S})$ is the state transition function, $R : \mathcal{S} \times \mathcal{A} \times \mathcal{S} \rightarrow \mathbb{R}$ is the reward function, $d : \mathcal{S} \rightarrow \text{Dist}(\mathcal{S})$ is the initial state distribution, and $\gamma \in (0, 1]$ is the discount factor. An agent needs to learn a policy $\pi : \mathcal{S} \rightarrow \text{Dist}(\mathcal{A})$ that maximizes the value, *i.e.* the expected discounted cumulative reward $\mathbb{E}_{\pi,P}[\sum_{t=0}^{T_\perp} \gamma^t R(S_t, A_t, S_{t+1})|S_0 \sim d]$, where $T_\perp$ denotes the timestep when the episode terminates. Often, environments are partially observable, which means, instead of a state, the agent receives an observation $x_{t+1}$, with which the agent needs to infer the state from the history.

**Target-Directed RL** offers a perspective to better identify delusions, inspired by delusion-related research in psychiatry (Kiran & Chaudhury, 2009). It abstracts existing methods that belong to the intersection of decision-time planning and generalized goal-conditioned RL (Zadem et al., 2024). The framework emphasizes an algorithmic design, where a *generator* proposes candidate targets, and an (optional) *estimator* evaluates them and select one target for policies to execute (Dayan & Hinton, 1992). The two components correspond to the belief formation and belief evaluation systems, respectively, whose incoordination causes delusions in the human brain, as shown in Fig. 1.

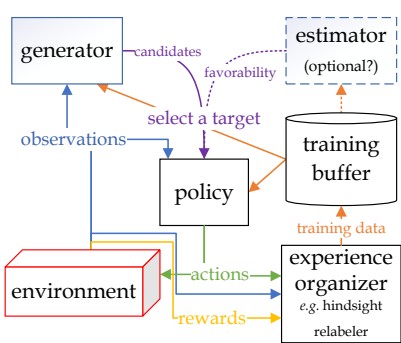

Figure 1: **Target-Directed Framework**: *A generator proposes candidate target(s). An estimator can be used to evaluate and select the favorable among candidates.*

**Source-Target Pairs & Hindsight Relabeling** Essentially, target-directed agents are trained on "source-target pairs", which are organized training data that could let the agents learn about the relationship between the source state and the target state. Training target-directed agents with only contemporary targets (the ones being followed) can lead to poor performance (Dai et al., 2021), since contemporary targets may be low-quality, hard to achieve, or lack in diversity (Moro et al., 2022; Davchev et al., 2021). Hindsight Experience Replay (HER) is proposed to be an experience organization mechanism, transforming collected experiences into relabeled transitions augmented with alternative targets (Andrychowicz et al., 2017). HER augments a transition $\langle x_t, a_t, r_{t+1}, x_{t+1} \rangle$ with an additional observation $x^\odot$ (or its encoding), the relabeled target, creating a *source-target pair*, with one decision point for the current step and another for the relabeled target. **Relabeling strategies**, which correspond to how $x^\odot$ is selected, are critical for the performance of HER-trained agents (Shams & Fevens, 2022). Most popular choices are *trajectory-level*, meaning $x^\odot$ comes from the same trajectory as $x_t$. These include "future", where $x^\odot = x_{t'}$ with $t' > t$, and "episode", with $0 \le t' \le T_\perp$.

**SSM** In existing works, failure modes of target-directed agents are often overlooked, possibly due to a lack of access to ground truths in benchmark environments. To identify the causes, and provide intuitive examples, we craft a set of fully-observable environments based on the MiniGrid-BabyAI framework (Chevalier-Boisvert et al., 2018b; Hui et al., 2020), named SwordShieldMonster (SSM for short). In SSM, the agent moves one step at a time in 4 absolute directions to navigate fields with episode-terminating, randomly placed lava traps. The density of traps is specified by a *difficulty* parameter $\delta$, while guaranteeing a viable path to success, *i.e.* acquiring a sparse terminal reward. Agents must collect a sword and a shield, both also randomly placed, before reaching the "monster" guarding the reward. Reaching the monster prematurely ends the episode. Visualizations of SSM are in Fig. 2, with examples of delusional behaviors discussed later. The sword and shield are picked up by moving to their respective grid cells, and agents cannot drop them. Such design introduces temporary unreachability in the state structure, as non-terminal states are not fully traversable from one to another. Semantically, this segments SSM states into 4 equivalence classes, formed by 2 binary indicators: the agent's possession of the sword and shield. For example, $\langle 0, 1 \rangle$ denotes sword not acquired, shield acquired". Despite SSM's simplicity in terms of observation space, a similar set of environments much simpler than SSM was already shown to be a significant challenge for state-of-the-art methods due to the OOD challenges (Zhao et al., 2024).

## 3 DELUSIONS IN TARGET-DIRECTED RL

To ultimately enable the agents to address delusions autonomously, we first need to identify where the problems reside. We first look into the generator to find its contributions to the emergence of delusional behaviors, and then locate delusions and their causes in the estimator.

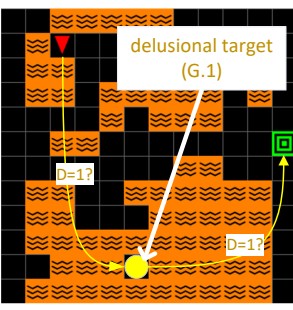 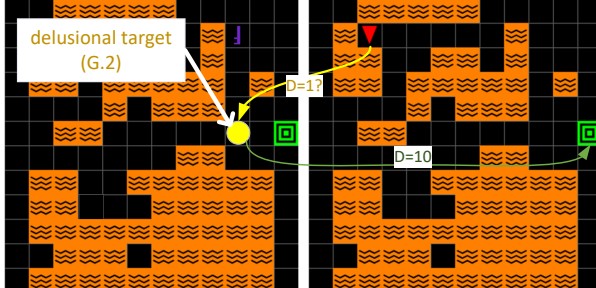

(a) E.1: *both planned steps are delusional*    (b) E.2 (current state in $\langle 1, 1 \rangle$, target in $\langle 0, 0 \rangle$): *first planned step is delusional*

Figure 2: **Delusional Behaviors in SSM**: *The agent location is marked as a red triangle, lava traps' as orange squares, and monsters' as green loops. In both cases, the estimators, lacking understanding of the problematic targets, fail to estimate the reachability of the problematic targets proposed at decision-time (yellow dots), leading to the delusions that shorter paths to the task goal exist through the problematic targets.*

### 3.1 GENERATOR'S ROLE IN DELUSIONAL BEHAVIORS: PROPOSING BAD TARGETS

Most learned generators are subjected to unwanted generalization, thus inevitably generate problematic targets (unreachable or unsafe), a behavior characterized as *hallucination* (Jafferjee et al., 2020). While hallucination can be reduced to a degree, they are generally unavoidable, especially in OOD scenarios. Despite that not all delusions are caused by problematic targets, generator hallucinations pose a major safety risk (Liu et al., 2022; Bengio et al., 2024).

Generators are implemented in various ways, thus it is ineffective to enumerate the detailed causes of problematic target generation. We skip these discussions and directly focus on a categorization of problematic targets, based on an often neglected perspective - the temporal relationship inside the source-target pairs. This leads to the following two *disjoint* types of problematic targets:

#### 3.1.1 TYPE G.1 - NONEXISTENT

The first type of problematic targets includes those that do not *correspond* to valid states in the task MDP. They can be classified into: 1) **Invalid**: semantically invalid targets for the task, such as a target SSM observation generated without the agent's location; 2) **Impossible**: semantically valid but unreachable targets, *e.g.* an SSM target cell surrounded by lava traps, (such as in Fig. 2 a). G.1 targets only cause delusional behavior if the estimator produces matching E.1 delusions, to be discussed later. Imperfectly generated targets *corresponding* to valid states may not be problematic, as estimators may filter out imperfections, possibly using a state encoder.

#### 3.1.2 TYPE G.2 - TEMPORARILY UNREACHABLE

The second type includes targets that correspond to valid states in the task MDP, but *cannot be fulfilled from the current state*. Unlike G.1, these G.2 targets are only temporarily inappropriate and could be perfectly good targets if the agent were in a different state. For the time-dependency, G.2 can be easily overlooked. There are some notable subtypes, such as: 1) **Irreversible**: targets once reachable but are now blocked due to a past transition. For instance, after acquiring the sword in SSM, the agent transitions from class $\langle 0, 0 \rangle$ to $\langle 1, 0 \rangle$, sealing off access to $\langle 0, 0 \rangle$ or $\langle 0, 1 \rangle$; 2) **Segregated**: targets that can only be reached from specific initial states that are not available to the agent. For example, an agent spawned in class $\langle 1, 0 \rangle$ cannot reach states in $\langle 0, 0 \rangle$ or $\langle 0, 1 \rangle$.

G.2 targets are often overlooked in literature, since hallucinations are mostly discussed in contexts without temporal progression, and also they do not exist in all MDPs. G.2 can appear more frequently if we do not avoid training the generators with them. For example, a conditional target

generator which learns from "episode" will be more likely to produce G.2 targets (compared to "future"), such as the one in Fig. 2 b). This applies to training beyond HER as well.

Problematic targets such as G.1 and G.2 can cause delusional behaviors of chasing unreachable targets, if they are favored during decision-time.

## 3.2 Estimator's Role in Delusional Behaviors: Bad Evaluation

An estimator evaluates the favorability of proposed targets, based on some relationship between the current states (sources) and the candidates (targets), which often involve assessing the implementability of targets, represented as, *e.g.*, distances, *etc.* Acting as a firewall, the estimator of a target-directed agent can enable the selection among the candidates, enabling the potential to filter out hallucinated targets. Agents without estimators, *e.g.* Director (Hafner et al., 2022), must accept proposed targets unconditionally, and thus are at significant risk of chasing problematic targets.

Incorporating an estimator only delegates the responsibility of avoiding delusional behavior from the generator to the estimator, as the estimators' own delusional beliefs about targets can directly cause delusional behaviors. A lack of either of the following two necessary aspects results in blind spots of knowledge about targets, *i.e.* delusions regarding both delusional or *non-delusional* targets.

**Update Rules**: we expect estimators to learn to reject problematic, undesirable or unsafe targets, by evaluating them with low favorability. However, such expectation may not be achieved, if the estimators' update rules do not have the characteristic that *(continuously) punishes unachieved / unsafe targets*, when these targets are sampled during training for the estimator to learn from.

**Training Data** Even with proper update rules, delusions can still arise from the lack of exposure to data counteracting the estimators' false beliefs. This is often caused by improper experience organization mechanisms, responsible for transforming the experienced interactions with environments into training data, as shown in Fig. 1. Despite these mechanisms like HER can lead to more diversity in training data, they are often accompanied by delusions: **1)** Certain relabeling strategies naturally cause delusions. For instance, "future" only relabels with future observations, thus only exposes a learner to future reachable targets, leaving the estimator to guess when a "past" target is proposed at decision-time; **2)** Trajectory-level relabeling can also be problematic. Short trajectories, common in many training procedures, only cover limited portions of the state space and prevent estimators from learning about distant targets, leaving estimators to guess when a distant target is generated at decision-time. Short trajectories can be a product of experimental design (initial state distributions, Maximum Episode Lengths, MELs (Erraqabi et al., 2021)) or innate environmental characteristics (density of terminal states). It is worth noting that delusions caused by improper training data (organizing mechanisms) also extend to cases like few-shot generalization.

With these necessary conditions, we identify types of estimator delusions based on the targets they evaluate. We use the identifiers E.0, E.1, and E.2 for their resulting delusional behaviors.

### 3.2.1 Type E.0 - Misevaluating Non-Delusional Targets

The standalone type of delusions describes false estimations about *non-delusional* targets. E.0 delusions can lead to undesirable or delusional targets being favored and thus can hurt generalization.

### 3.2.2 Type E.1 - Misevaluating G.1 Targets

This type of delusion appears when an estimator misevaluates the favorability of a G.1 target. If effective learning rules are present, this can still be caused by the lack of necessary training data (*e.g.*, relabeled transitions with G.1 targets in buffer). In Fig. 2 a), an example is visualized. Note that the resulting E.1 behaviors, *i.e.* those chasing selected G.1 targets, can be potentially catastrophic if the G.1 targets are beyond safety constraints.

### 3.2.3 Type E.2 - Misevaluating G.2 Targets

Similar to E.1, in Fig. 2 b), an example of delusional behavior is visualized, a result of a combination of G.2 (temporarily unreachable) and E.2. E.2 behaviors can hurt the agents' generalization abilities.

## 4 Addressing Delusions in Target-Directed Agents

Having identified delusions, taking advantage of the shared reliance on source-target pairs, we can develop strategies applicable to general target-directed agents coming from various training procedures. Then, we provide examples on how to materialize such strategies in agents trained with HER, because of its popularity, its direct association with source-target pairs and the fact that one can trivially transform HER-based training into losses established over two sampled transitions from ordinary experience replays.

Without the loss of generality, assume we are dealing with a hypothetical target-directed agent with a dual-component architecture as in Fig. 1, learning both components *exclusively* from hindsight-relabeled transitions. The generator proposes targets corresponding to potentially distant states, and the estimator learns the favorability of targets by (continuously) punishing the unachieved or unsafe targets sampled during training. Note that even for methods whose generators are not trained with HER, our estimator-focused strategies still apply.

As discussed earlier, to address delusions, we need effective *update rules* and *training data*. By assuming this framework, we skip discussing update rules, as they depend on the specific designs of the chosen target-directed agents. Our emphasis on the training data shines light on the fact that most existing RL agents only learn from experienced data, while addressing delusions requires learning from targets that can never be experienced. This corresponds to a discrepancy between (most existing) target-directed agents' behaviors and training: at decision-time, targets outside experience can be proposed; while during training, only experienced targets are learned from.

### 4.1 Assistive Strategies for Addressing Delusions

We first introduce two ideas to improve training data distributions, which can then be materialized as two hindsight relabeling strategies for HER. These ideas seek to expand the support of the training data distribution, to include those source-target combinations that the agent could never experience, *i.e.* those involving G.1 and G.2 targets. Since our proposed ideas do not rely on additional assumptions, they should be expected to be applicable generally. Additionally, with convergent learning rules (provided by the target-directed framework that these strategies are applied to), the strategies should also lead to the correct estimation of G.1 and G.2 targets.

Later, we compare the ideas behind all HER strategies, including the existing and the newly proposed, and study how the two groups can be combined to accommodate the learning needs.

#### 4.1.1 "generate": Let Estimators Learn About Candidates (to be generated)

The first strategy, named "generate", is to *let the estimator learn about targets that could be proposed at decision time*, *s.t.* it could figure out preemptively that problematic targets are not favorable.

Zhao et al. (2024) identified delusional behaviors resulted from E.1 delusions, in the language of this paper, and proposed to train the estimator additionally with candidate targets proposed by the generator. With HER, we can transform this auxiliary loss into a Just-In-Time (JIT) HER strategy that, as a transition is sampled for training, relabels it with a proposed target by the generator. We can expect "generate" to be effective for training estimators, as the estimators will get exposed to all kinds of problematic targets that the generator could offer: When a generator learns to generate mostly viable targets, "generate" will be primarily effective at dealing with the E.0 and E.1 (G.1s are unlikely to be eradicated). "generate" requires the use of the generator, thus it incurs additional computational burden, depending on the complexity of target generation processes.

Adapting "generate" beyond HER should be straightforward: whenever a target is involved during estimator learning, we may replace it with generated targets. "generate" comes at a cost of extra computation, to which attention may be required in real-world applications with speed demands.

#### 4.1.2 "pertask": Let Estimators Learn About Experienced Targets Outside the Episode

The second strategy, namely "pertask", is to *expose the estimator to all targets experienced before, s.t.* it could figure out that some previously achieved targets are unreachable from the current state.

We materialize "pertask" into an assistive atomic strategy that relabels transitions with observations from the same training task, sampled across the entire memory. Importantly, "pertask" brings exposure to the estimator to learn against E.2 delusions and to the long-distance source-target pairs against E.0 caused by short trajectories. Take SSM as an intuitive example, a current state in situation $\langle 1, 0 \rangle$ in the current episode can now be paired with a target in $\langle 0, 1 \rangle$ in another episode, *s.t.* the estimator learns such G.2 target is unreachable, as shown in Fig. 4 (in Appendix).

It is worth noting that "pertask" also biases the training data distribution, making the agent spread out its efforts into learning the source-target pairs potentially far away from each other. Despite increasing training data diversity, long-distance pairs are less likely to contribute to better decision-making compared to the shorter-distance in-episode ones offered by "episode", as shown later.

Adapting "pertask" beyond HER requires algorithmic designs recording all past observations.

Both strategies help the estimator figure out the features shared by problematic targets, *s.t.* OOD delusions can also be identified. Note that "pertask" cannot be used to address E.1 delusions. However, "generate" can be used for E.2 if the generator generates the respective problematic targets.

## 4.2 MIXTURES

Creating a mixture of sources of training data increases the diversity of source-target combinations. For HER specifically, each atomic strategy, enumerated in Tab. 1, exhibits a tradeoff in estimation accuracy among all sorts of source-target pairs, including short-distance and long-distance ones involving only valid targets, and those involving problematic targets.

A mixture of more-than-one atomic strategies in certain proportions while relabeling (Nasiriany et al., 2019; Yang et al., 2021a), can be used to achieve a tradeoff in HER-based training, *s.t.* the shortcomings of each atomic strategy are mitigated by the introduction of others.

| Strategies | Advantages | Disadvantages | Gist |
|---|---|---|---|
| "episode" | Efficient for estimator to learn close-proximity relationships | Can cause G.2 targets when used to train generators; When used exclusively to train estimator, 1) cannot handle E.2; 2) prone to E.0 - cannot learn well from short trajectories | Creates training data with source-target pairs sampled from the same episodes |
| "future" | Can be used to learn a conditional generator with temporal abstractions | In addition to the shortcomings of "episode" (those for estimators only), this additionally causes E.0 when used as the exclusive strategy for estimator training | Creates training data with temporally ordered source-target pairs from the same episodes |
| "generate" | Addresses E.1 with data diversity (also E.2 when generator produces G.2) | Relies on the generator with additional computational costs; Potentially low efficiency in learning non-delusional relationships. | Augments training data to include candidate targets proposed at decision time |
| "pertask" | Addresses estimator delusions (E.2 & E.0 for long-distance pairs) | Can cause extensive G.2 targets if used to train generators; low efficiency in learning close-proximity source-target relationships | Augments training data to include targets that were experienced |

Table 1: **Hindsight Relabeling Strategies & The Ideas Behind**: *"episode" and "future" are widely used as they increase sample efficiency in non-delusional cases significantly; "generate" and "pertask", proposed in this paper, are effective against delusions, useful in specific scenarios.*

When the training budget is fixed, *i.e.*, training frequency, batch sizes, *etc.*, stay unchanged, the mixing proportions of strategies pose a tradeoff to the investment towards different kinds of source-target pairs, and the resulting accuracies. In experiments, we show that, assisting "episode" with "generate" and "pertask" often results in better performance in estimator training, striking a balance between the investment of training budgets in non-delusional and delusional estimations.

Mixtures of multiple relabeling strategies can also be extended to other training procedures, and can also be implemented by independent training losses based on different sources.

## 4.3 HYBRID STRATEGIES - A 2-SLOTTED APPROACH

While an estimator needs the exposure to problematic targets to counteract delusions, a generator benefits from learning to generate only useful targets, and should avoid exposure to problematic targets. Thus, generators and estimators often have conflicting needs during training.

Each source of training data has its own biases. Atomic hindsight relabeling strategies in HER, as shown in Tab. 1, can simultaneously help one component while hindering another. Instead of trying to tradeoff the needs of both components with a single source of training data, we propose a hybrid (2-slotted) approach, allowing the generator and estimator to receive data tailored to their needs, through two independent relabeling processes. This combination of 2 slots and mixtures can produce various hybrid strategies, as demonstrated in Sec. 5.4.

Separating training data for the generator and estimator based on their needs is straightforward in training procedures beyond HER.

*In the Appendix, we provide summaries for the design of mitigation strategies on agents beyond HER.*

## 5 EXPERIMENTS

To investigate the effectiveness of our strategies against delusions and how they help target-directed agents achieve better generalization, we implement a combination of *environments* with clearly defined delusional cases, as well as *methods* whose estimators can be easily analyzed. This leads to our 4 sets of experiments, coming from a combination of 2 environments (one dominantly haunted by G.1, and another by G.2) and 2 target-directed frameworks. *Due to page limit, 3 out of 4 sets of experiments are presented in the Appendix. We summarize those at the end of this section.*

### 5.1 ENVIRONMENTS & SETTINGS

For environments, we favor tasks where the delusions are present, dangerous and intuitive to inspect, for which the introduced SSM offers us clear advantage. *Introduction and the results on another environment are presented in the Appendix.*

For each training seed, we sample and preserve 50 training tasks of size $12 \times 12$ and difficulty $\delta = 0.4$. All agents are trained for $1.5 \times 10^6$ interactions by randomly selecting one of the 50 frozen tasks for each training episode. To speed up training, we make the initial state distributions span all the non-terminal states in each training task. This change increases risks of E.2, due to the presence of dense episode-terminating lava traps and relatively short MELs (128 for SSM).

### 5.2 EVALUATIVE CRITERIA

We use the following criteria to investigate the changes in agent's estimations and behaviors:

**Estimation Error - E.0, E.1, E.2 & Non-Delusional**: At each evaluation point, we solve the ground truth distances between states and inspect the agents' estimation errors. The errors are split based on the source-target pairs, into those involving problematic targets, *i.e.*, E.1 and E.2, and those involving only valid targets, which includes the case of E.0. The estimation errors show the internal degree of delusions about targets within the estimators.

**Delusional Behavior Frequencies**: We also monitor the frequency of a problematic target (G.1 or G.2) being chosen by the agents, as the result of their delusions in decision-time planning. The frequencies demonstrate how often delusional targets are being favored because of delusions.

**Improvement on OOD Generalization Performance**: We analyze the changes in agents' OOD generalization performance, due to the strategies introduced to handle delusions. The evaluation tasks are sampled from a gradient of OOD difficulties - 0.25, 0.35, 0.45 and 0.55. For aggregated OOD performance, such as in Fig. 3 g), we sample 20 tasks from each of the 4 OOD difficulties, and combine the performance across all 80 episodes, which have a mean difficulty matching the training tasks. By comparing the resulted performances of atomic strategies with those of the newly proposed hybrids, we can also have an intuitive grasp of the "delusion gap", which is the amount of performance degradation caused by delusions. To maximize difficulty, the initial state is fixed in each novel evaluation task instance: the agents are not only spawned to be at the furthest side of the monster, but also in semantic class $\langle 0, 0 \rangle$, *i.e.*, without the sword nor the shield in hand.

### 5.3 Methods

We seek to demonstrate the generality of our strategies by applying them onto different methods with very different behaviors. Thus, we adapted the following target-directed methods:

Skipper: generates candidate target states that, together with the current state, form a directed graph at decision-time. The states act as vertices and the edges are pairwise estimations of cumulative rewards and discounts (interchangeable with distances), under its evolving policy. A target state is chosen after applying *value iteration (VI), i.e.* the favorability of targets are defined by the $Q$ values of the planned paths (Zhao et al., 2024).

LEAP **(in Appendix)**: Instead of VI, LEAP uses an evolutionary algorithm to evolve sequences of subgoals leading to the task goal, with distances as fitness. The immediate subgoal of the elitest sequence is chosen to be the target, and used to condition a lower-level policy. LEAP is prone to delusions, since one problematic subgoal can destroy a whole sequence (Nasiriany et al., 2019).

For both agents, the generators and estimators are trained with HER, allowing easy switching of relabeling strategies for comparison. Roughly, both agents employ update rules that can reject wrong estimations with data exposure (more discussions in Appendix). Targets are acquired by encoding observations imagined by the generators, where G.1 & G.2 propositions are clearly identified.

### 5.4 Hindsight Relabeling Strategies

We focus on the following *baselines* for fairer comparison (excluding bad generator performance):

- **"F-E"**: "future" for generator, "episode" for estimator;
- **"F-P"**: "future" for generator, "pertask" for estimator;
- **"F-G"**: "future" for generator, with 100% chance using "generate" JIT on estimator;

Additionally, the following hybrid strategies are proposed and compared:

* **"F-(E+G)"**: A hybrid strategy mostly for E.1. "future" for generator, "episode" for estimator with 50% chance using "generate" JIT, resulting in a half-half mixture of "episode" & "generate";
* **"F-(E+P)"**: mostly for E.2. "future" for generator, half "episode" & half "pertask" for estimator;
* **"F-(E+P+G)"**: for both E.1 & E.2 behaviors. "future" for generator, mixture of 2/3 "episode" and 1/3 "pertask" for estimator, with 1/4 chance using "generate" JIT, resulting in a mixture of 50% "episode", 25% "pertask" and 25% "generate" for estimator;

### 5.5 Skipper on SSM (Set 1/4)

**Generators**: We first verify the generator's contributions to delusional behaviors. For the HER-trained generators, Fig. 3 **a)**, shows that none among "future", "episode" and "pertask" could significantly address G.1 target generation, validating our claims about the challenges of hallucination. While, Fig. 3 **e)** indicates that, "future" generates G.2 delusional targets significantly less frequently than "episode" and "pertask", as the other two wasted training budget on G.2 targets, especially "pertask" that brings in more problematic training samples from long distances. *For fairer comparison in the estimators, we only compare variants with "future" for the generator training, i.e., "F-*".*

**Estimators**: Now, we look into the degrees of delusions inside the learned estimators. Exclusive use of "episode" for estimators resulted in high accuracy in non-delusional cases (cases involving pairs of non-delusional states, "F-E" in Fig. 3 **d)**) for source-target pairs of both short and long distances, but low accuracies in E.1 and E.2 cases (Fig. 3 **b)** & **f)**). This shows the source of popularity of these baseline strategies in existing literature; Unsurprisingly, exclusive use of "pertask" results in significantly worse non-delusional short-distance estimates ("F-P" in **d)**), yet much better than "episode" ("F-E") and "future" ("F-F") in long-distance cases (indicating that trajectory-level "future" and "episode" lead to E.0 cases for longer-distance source-target pairs). This is likely because, without the presence of a backbone strategy such as "episode", the estimator wastes its budget learning long-distance non-delusional or E.2 cases; Demonstrating a similar tradeoff, for "generate" ("F-G"), high accuracy is achieved for E.1 related cases, at the sacrifice of E.2 and non-delusional cases; All 3 hybrid non-baseline strategies achieve good estimation accuracies in both

short- and long-distance non-delusional estimation, as shown in **d)**. At the same time, we can observe particularly significant improvement in accuracy in E.2 delusional estimates in **f)**, by "F-(E+P)" and "F-(E+P+G)", the hybrids assisted by "pertask". These indicate their effectiveness.

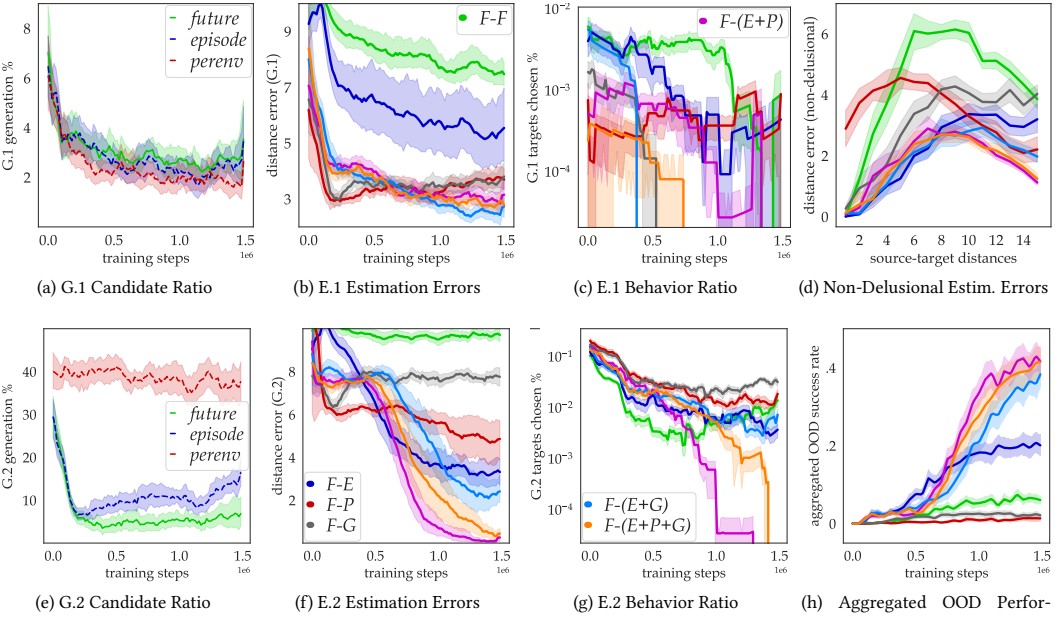

(a) G.1 Candidate Ratio  (b) E.1 Estimation Errors  (c) E.1 Behavior Ratio  (d) Non-Delusional Estim. Errors

(e) G.2 Candidate Ratio  (f) E.2 Estimation Errors  (g) E.2 Behavior Ratio  (h) Aggregated OOD Performance

Figure 3: Skipper **on** `SSM`: *each mean curve and CI (95% for all subfigures except **c**) & **g**), which used 50% due to the chaotic overlap) bar are established over* 20 *seed runs. Legends are shared among* **b,c,d,f,g,h**). **a-c)** *focus on E.1 delusions:* **a)** *Evolving ratio of G.1 targets among all candidates at each target selection, throughout training. Variants are differentiated only with the generator HER strategies;* **b)** *Evolving E.1 delusions measured by* $L_1$ *error in estimated distance throughout training. The distances to unreachable targets are clipped to be the maximum values that the estimator could output (*16 *in experiments). A more detailed breakdown of this error is in the Appendix;* **c)** *The curves represent the frequencies of choosing G.1 targets whenever a selection of targets is initiated;* **d)** *Final estimation accuracies of non-delusional source-target pairs after training completed, across a spectrum of ground truth distances. Both estimated and true distances are conditioned on the policies of the last timestep. Note that a part of the errors comes from E.0; Subfigures **e-g**) are the E.2-counterparts of **a-c**);* **h)** *Each data point represents OOD evaluation performance aggregated over* $4 \times 20$ *newly generated tasks, with mean difficulty matching training. The decomposed curves are presented in the Appendix.*

**Behaviors**: Finally, we examine how reducing delusions affect the agents' behaviors. For all compared variants, we can deduce from 3 that generally, less frequent G.2 generation (**e)**) and lower E.2 errors (**f)**) lead to less frequent delusional behaviors in **g)**, which in turn improves the OOD performance in **h)**; "episode" ("F-E") showed mediocre OOD performance (**h)**) for its good estimation accuracy in short-distance non-delusional cases, but was prone to delusional behaviors; While, despite that "pertask" ("F-P") showed decent accuracies in longer-distance non-delusional and E.2 cases (**d)** & **f)**), its low accuracy in short-distance non-delusional estimation (**d)**) devastated the baseline to the lowest performance in **h)**; Similarly, despite "generate"'s ("F-G"'s) effectiveness in addressing E.1-behaviors (**c)**), its resulting bad estimation accuracies in the non-delusional cases destroyed its OOD performance. In contrast, all 3 hybrids achieve significantly better OOD performance in **h)**. Aassisted by "pertask", "F-(E+P)" and "F-(E+P+G)" performed the best in reducing E.2 delusions (in **f)**) and consequently addressed the most delusional behaviors (in **g)**). This indicates a major presence of E.2 behaviors of Skipper's failure modes on `SSM`.

### 5.6  SUMMARY OF EXPERIMENTS

With the proposed strategies, we saw a reduction in delusions in terms of estimation errors and in delusional behaviors in both Skipper and LEAP (in Appendix, more prone to delusions), which led to better OOD generalization performance in 2 sets of environments, posing challenges of G.1 & G.2, respectively (more results in Appendix). All 4 sets of experiments align in terms of conclusions.

## 6 Related Works

**Target-Directed Agents.** Dual-component frameworks are explicitly investigated with temporal abstraction for purposes such as path planning (Nasiriany et al., 2019), OOD generalization (Zhao et al., 2024) and task decomposition (Nair & Finn, 2020; Zadem et al., 2024). Davchev et al. (2021) employs the similar framework to assist exploration, without using HER for training.

**Hindsight Relabeling** is crucial for enhancing *sample efficiency* in goal-directed RL (Andrychow-icz et al., 2017), as it enables learning from failed experiences (Dai et al., 2021). From the perspective of source-target pairs, improvements were proposed, including divergence maximization (Zhang & Stadie, 2022), addressing distributional change (Bai et al., 2023), multistep relabeling strategies (Yang et al., 2021b), prioritizing rarely-seen achieved transitions (Kuang et al., 2020), *etc.*

HER's success was centered on its performance gains via sample efficiency (in non-delusional cases), around which most follow-up works revolved as well. In reality, performance deficiencies of target-directed frameworks can be resulted from multiple factors, one of which is delusions. Shams & Fevens (2022) studied the performance of atomic strategies from the view of sample efficiency, without looking into the failure modes. Deshpande et al. (2018) detailed experimental techniques in sparse reward settings using "future". In (Yang et al., 2021a), a mixture strategy similar to "generate" improved non-delusional estimations, though the impact on delusions was not explored, possibly because of the lack of an appropriate environment. Nasiriany et al. (2019) used a mixture of up to 3 atomic strategies for producing effective training data in a single-task setting, while the contributions of the mixture to addressing delusions were not investigated. It is widely known that performance of existing HER-trained agents are often limited by their exclusive reliance on "future" or "episode" (He et al., 2020), whose delusions this paper intends to address.

**Delusions** in RL can stem from other improper designs. Lu et al. (2018) identified delusions caused by *the limitations of function approximators* for greedy policies in model-free RL; Often accompanying hallucinations, delusions can arise in non-target-directed frameworks as well. In an offline planning framework such as Dyna, hallucinations can irreversibly destabilize value estimations (Jafferjee et al., 2020). Di Langosco et al. (2022) classified goal misgeneralization, a class of delusional behaviors with which an agent competently pursues an undesired target leading in novel test situations. Zhao et al. (2024) gave first examples of delusional behaviors caused by improper hindsight relabeling. The perspective of delusions can often be overlooked because of the lack of estimators in target-directed methods, as discussions regarding hallucinations get magnified.

## 7 Empirical Guidelines

The following steps can be applied to target-directed methods (applicable also beyond HER):

1. Incorporate an estimator to enable rejection of unwanted targets;

2. Use proper update rules and try to maximize the diversity of the training data (against E.0);

3. Inspect the candidate targets if possible, as their low quality may indicate higher E.1 risks, which can be mitigated with "generate";

4. Analyze the state structure of the target tasks and identify temporary unreachabilities. These E.2 risks may indicate the effectiveness of "pertask".

## 8 Conclusion & Future Work

We investigated the causes of delusional behaviors, a shared failure mode of target-directed agents. When applying our mitigation strategies on HER, the resulting hybrid hindsight relabeling strategies flexibly satisfy the needs of the generator and the estimator, granting the agents the ability to address delusions autonomously and preemptively avoid delusional behaviors.

It is likely that we did not exhaustively identify all potential types of delusions in target-directed agents. We wish to continue the investigation into unidentified causes of failure modes and design more useful skill-learning agents.

## 9 REPRODUCIBILITY STATEMENT

The results presented in the experiments are **fully**-reproducible with the submitted source code.

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

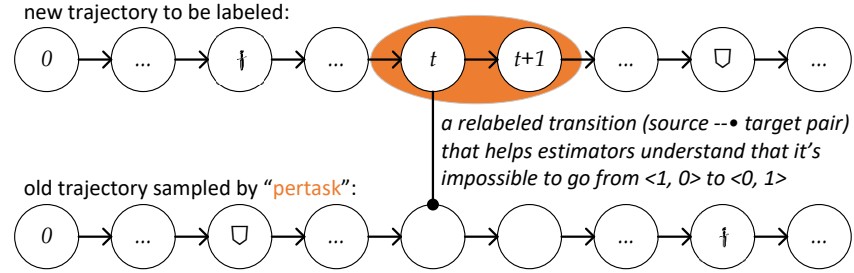

Figure 4: **An Example of How "pertask" Addresses E.2**: The new trajectory contains the events of acquiring the sword first and the shield later. While the old trajectory sampled by "pertask" acquired the shield first and then the sword. The acquisition of swords and shields are marked with icons on the corresponding states. when relabeling a transition in the new trajectory over timestep $t$ to $t + 1$ (in $\langle 1, 0 \rangle$), a target observation in an existing trajectory (in $\langle 0, 1 \rangle$) can be paired to create a source-target pair that can make an agent realize the pair's un-implementability, therefore reducing E.2 delusions about the G.2 target.

## A SUMMARIES

For better understanding of the contributions of this paper, we provide some brief summaries about the identified delusions / delusional behaviors and the corresponding mitigation strategies. For generator delusions, please check Tab. 2. For estimator delusions, please check Tab. 3. We hope that these could inspire the application on more target-directed methods.

| Types | Causes | Solutions |
|---|---|---|
| G.1 | The use of neural networks in the generator architecture makes G.1 targets almost inevitable | Can cause G.2 targets when used to train generators; No general solution for this type |
| G.2 | Problematic training procedures that did not isolate the generators from temporarily unreachable targets | Isolate generator from learning temporarily unreachable states |

Table 2: **Causes & Solutions for Generator Delusions**

| Causes | Explanations | Solutions |
|---|---|---|
| Inappropriate Update Rules | Some update rules do not lead to punishment of the favorability of unachieved targets | Adjust the learning rules to continuously punish the unachieved targets |
| Bad Training Data Distributions | the collected experience was not organized in a way that produced sufficiently diverse source-target distributions to make the agent realize, via appropriate update rules, that wrong beliefs are wrong | Augment the training data with source-target pairs involving targets that could participate in planning, but will never be experienced, *i.e.* G.1 and G.2 |

Table 3: **Causes & Solutions for Estimator Delusions**

## B EXPERIMENTS

### B.1 SKIPPER ON SSM (SET 1 / 4, CONT.)

#### B.1.1 BREAKDOWN OF E.1 ERRORS

Skipper utilizes policy evaluation based on Temporal Difference updates to learn the cumulative rewards and discounts along the way from one state to another. These recursive update rules based

on state-action (Q) values enable the effective learning of estimates conditioned on the evolving policy within the trust regions. However, they are constrained in the sense that they can only learn the unfavorability of hallucinated targets *from* non-delusional states, not the other way around, if the considerate training procedure is present to provide samples against the delusions. This is particularly problematic for cases involving G.1 targets, but not those with G.2, since there exist source-target pairs in ER where the generated G.2 targets serve as sources.

Fig. 5 provides a breakdown of the E.1 errors, into cases depending on if the G.1 targets serve as the source or the target in the corresponding relationship estimation on the source-target pairs. From the accuracy gaps between pairs of variants in each case, we can see that, with our proposed hybrid strategies, Skipper can effectively deal with case **e)**, therefore leading to better accuracy in case **c)** and **a)**.

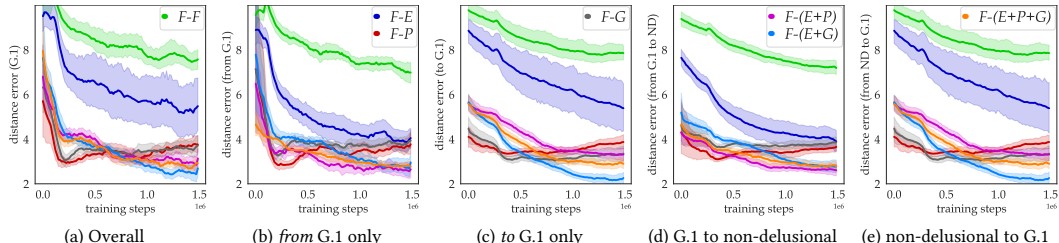

(a) Overall    (b) *from* G.1 only    (c) *to* G.1 only    (d) G.1 to non-delusional    (e) non-delusional to G.1

Figure 5: **Breakdown of E.1 Errors of** Skipper **on SSM**: each mean curve and CI (95%) bar are over 20 seed runs. In each subfigure, the evolution of $L_1$ errors against the ground truth is presented, with each data point computed based on the evolving policy.

### B.1.2 Breakdown of Task Performance

In Fig. 6, we present the evolution of Skipper variants' performance on the training tasks as well as the OOD evaluation tasks throughout the training process. Note that Fig. 3 **h)** is an aggregation of performances in Fig. 6 **b-e)**.

From the performance advantages of the hybrid variants (in both training and evaluation tasks), we can see that learning to address delusions during training brings better understanding for novel situations posed in OOD tasks.

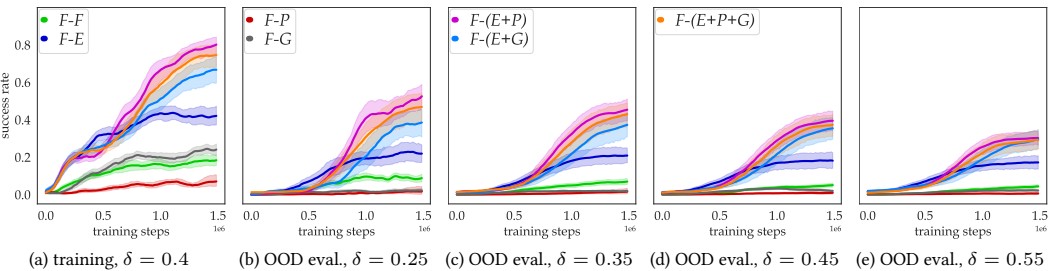

(a) training, $\delta = 0.4$    (b) OOD eval., $\delta = 0.25$    (c) OOD eval., $\delta = 0.35$    (d) OOD eval., $\delta = 0.45$    (e) OOD eval., $\delta = 0.55$

Figure 6: **Evolution of Performances of** Skipper **Variants on SSM**: each mean curve and CI (95%) bar are over 20 seed runs.

### B.2 Skipper on RDS (Set 2 / 4)

The second environment we employ is `RandDistShift`, abbreviated as `RDS`. `RDS` was originally proposed in Zhao et al. (2021) as a variant of the counterparts in the MiniGrid Baby-AI platform (Chevalier-Boisvert et al., 2018a;b; Hui et al., 2020) and then later used as the experimental backbone in Zhao et al. (2024). We can view `RDS` as a simpler version of `SSM`, where everything happens in semantic class $\langle 1, 1 \rangle$, *i.e.*, agents always spawn with the sword and the shield in hand, thus can acquire the terminal sparse reward by simply navigating to the goal. `RDS` instances thus have

smaller state spaces than its `SSM` counterparts. The most important difference, in the views of this paper, is that `RDS` removed the challenges introduced by temporary unreachabilities. This means that G.2 and E.2 are no longer relevant, shifting the dominance towards G.1 + E.1 combination. Using `RDS` not only showcase the performance of the proposed strategies on a controlled environment with G.1 + E.1 dominance, contrasting the G.2 + E.2 dominance of `SSM`, it also can be used to validate the performance of our adapted agents, on an environment where previous benchmarks exist.

We present Skipper's evaluative curves in Fig. 7.

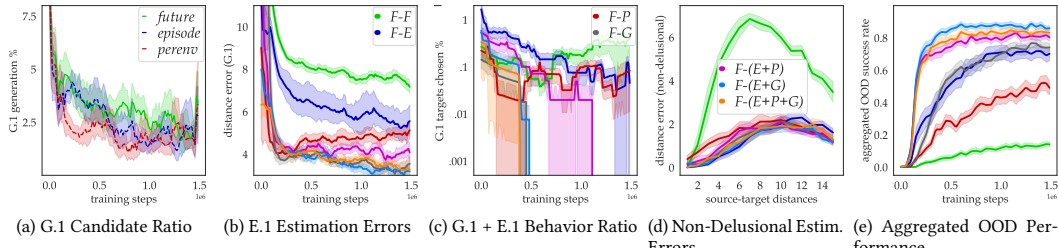

(a) G.1 Candidate Ratio  (b) E.1 Estimation Errors  (c) G.1 + E.1 Behavior Ratio  (d) Non-Delusional Estim. Errors  (e) Aggregated OOD Performance

Figure 7: Skipper **on RDS**: each mean curve and CI (95%) bar are over 20 seed runs. Subfigures **b-e)** share the same legends. Subfigures **a-c)** focus on G.1 & E.1 related delusions: **a)** Evolving ratio of G.1 among targets at each candidate selection throughout the training process is presented; **b)** E.1 delusions in terms of $L_1$ error in estimated distance is visualized, throughout the training process. **c)** The curves represent the frequencies of choosing G.1 targets whenever a selection of targets is initiated; **d)** The final estimation accuracies of non-delusional source-target pairs after training completed, across a spectrum of ground truth distances. In this figure, both distances (estimation and ground truth) are conditioned on the final version of the evolving policies; The state structure of `RDS` does not permit G.2 targets and the corresponding E.2 delusions; **e)** Each data point represents OOD evaluation performance aggregated over $4 \times 20$ newly generated tasks, with mean difficulty matching the training tasks.

From Fig. 7 **e)**, we can see that, probably because of the lack of dominant G.2 + E.2 cases, the OOD performance of even the most basic "episode" variant is high, despite the hybrid variants perform even better. "F-(E+G)", *i.e.* the hybrid with the most investment in "generate" (aiming at E.1), performs the best both in terms of E.1 delusion suppression (**b)**), and OOD generalization (**e)**), as expected. In `RDS`, the short-distance non-delusional estimation accuracy as well as the OOD performance of "F-P" are not as bad as in `SSM`. This is possibly due to the fact that `RDS` has much smaller state spaces, where "episode" and "pertask" produce more similar results (than in large state spaces of `SSM`).

### B.2.1 BREAKDOWN OF E.1 ERRORS

Similarly, for `RDS`, we provide a breakdown of E.1 errors for the Skipper variants in Fig. 8. Similar observations can be made as for `SSM`.

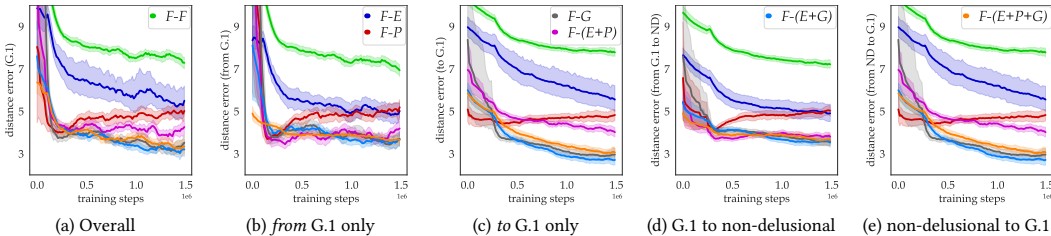

(a) Overall  (b) *from* G.1 only  (c) *to* G.1 only  (d) G.1 to non-delusional  (e) non-delusional to G.1

Figure 8: **Breakdown of E.1 Errors of** Skipper **on RDS**: each mean curve and CI (95%) bar are over 20 seed runs. In each subfigure, the evolution of $L_1$ errors against the ground truth is presented, with each data point computed based on the evolving policy.

### B.2.2 Breakdown of Task Performance

In Fig. 9, we present the evolution of Skipper variants' performance on the training tasks as well as the OOD evaluation tasks throughout the training process. Note that Fig. 7 **e)** is an aggregation of performances in Fig. 9 **b-e)**.

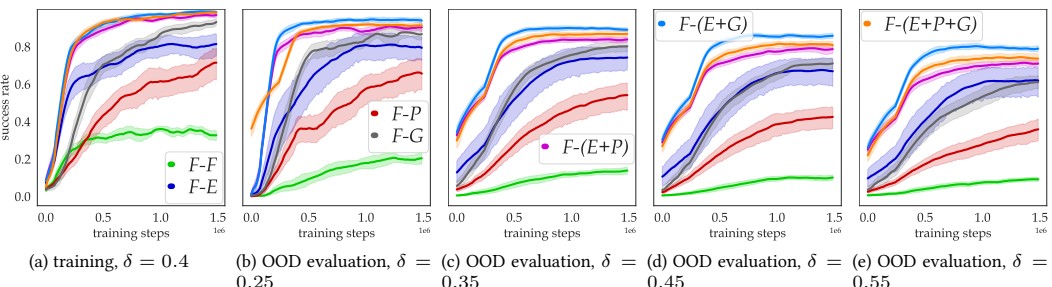

(a) training, $\delta = 0.4$    (b) OOD evaluation, $\delta =$ 0.25    (c) OOD evaluation, $\delta =$ 0.35    (d) OOD evaluation, $\delta =$ 0.45    (e) OOD evaluation, $\delta =$ 0.55

Figure 9: **Evolution of Performances of** Skipper **Variants on RDS**: each mean curve and CI (95%) bar are over 20 seed runs.

### B.3 LEAP on SSM (Set 3 / 4)

The third set of experiments, similar to the previous two, is a comparative study for LEAP variants' performance on SSM. LEAP is different from Skipper, as its decision-time planning process constructs a singular sequence of subgoals leading to the task goal. Due to a lack of backup subgoals, even if one among them is problematic, the whole resulting plan would be delusional, making LEAP much more prone to failures compared to Skipper, where candidate targets can still be reused if deviation from the original plan occurred.

SSM has a relatively large state space that requires more intermediate subgoals for LEAP's plans. However, an increment of the number of subgoals also dramatically increases the frequencies of delusional plans, damaging the agents' performance. Because of this, our experimental results of LEAP on SSM with size $12 \times 12$ became difficult to analyze because of the rampant failures. We chose instead to present the results on SSM with size $8 \times 8$ here.

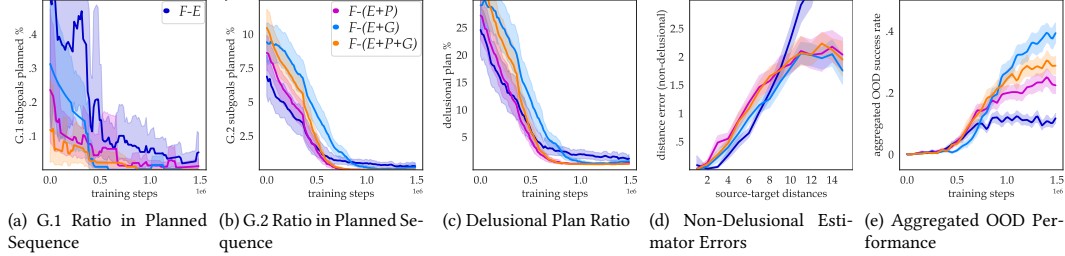

(a) G.1 Ratio in Planned Sequence    (b) G.2 Ratio in Planned Sequence    (c) Delusional Plan Ratio    (d) Non-Delusional Estimator Errors    (e) Aggregated OOD Performance

Figure 10: LEAP **on SSM**: each mean curve and CI (95%) bar are over 20 seed runs. **a)** Ratio of G.1 subgoals among the evolved sequence; **b)** Ratio of G.2 subgoals in the evolved sequence; **c)** Ratio of evolved sequences containing at least one G.1 or G.2 target; **d)** The final estimation accuracies of non-delusional source-target pairs after training completed, across a spectrum of ground truth distances. In this figure, both distances (estimation and ground truth) are conditioned on the final version of the evolving policies; **e)** Each data point represents OOD evaluation performance aggregated over $4 \times 20$ newly generated tasks, with mean difficulty matching the training tasks.

For LEAP, we use some different metrics to analyze the effectiveness of the proposed strategies in addressing delusions. This is because, if LEAP's estimator successfully addressed delusions and learned not to favor the problematic targets (G.1 and G.2), then they will not be selected in the evolved elitest sequence of subgoals. This makes it inconvenient for us to use the distance error

in the delusional source-target pairs during decision-time as a metric to analyze the reduction of delusional estimates, because of their growing scarcity.

As we can see from Fig. 10, similar arguments about the effectiveness of the proposed hybrid strategies can be made, to those with Skipper. The hybrids with more investment in addressing E.1, *i.e.*, "F-(E+G)" and "F-(E+P+G)", exhibit the lowest E.1 errors (**a**)). Similarly, "F-(E+P)" and "F-(E+P+G)" achieve the lowest E.2 errors (**b**)). In **e**), we see that the 3 hybrid variants achieve better OOD performance than the baseline "F-E". Specifically, "F-(E+G)" achieved the best performance. This is likely because that it induced the highest sample efficiency in terms of learning the estimations between non-delusional subgoals, as shown in **d**). Assistive strategies such as "generate" and "pertask" do not only induce problematic targets, but also non-delusional ones that can shift the training distribution towards higher sample efficiencies in the traditional sense.

### B.3.1 BREAKDOWN OF TASK PERFORMANCE

In Fig. 11, we present the evolution of LEAP variants' performance on the training tasks as well as the OOD evaluation tasks throughout the training process. Note that Fig. 10 **e**) is an aggregation of performances in Fig. 11 **b-e**).

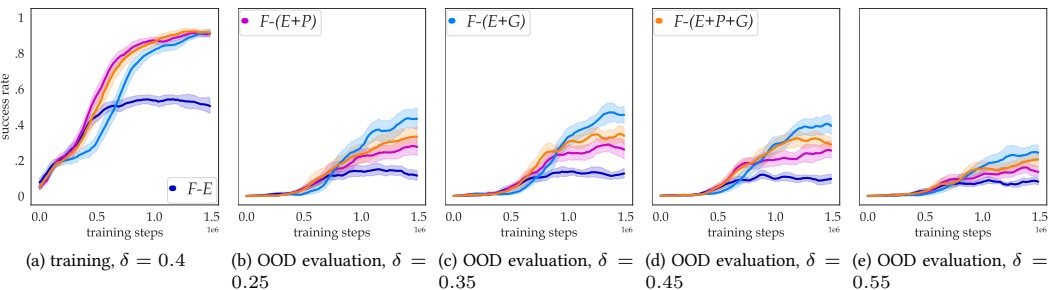

(a) training, $\delta = 0.4$    (b) OOD evaluation, $\delta = 0.25$    (c) OOD evaluation, $\delta = 0.35$    (d) OOD evaluation, $\delta = 0.45$    (e) OOD evaluation, $\delta = 0.55$

Figure 11: **Evolution of Performances of** LEAP **Variants on SSM**: each mean curve and CI (95%) bar are over 20 seed runs.

### B.4 LEAP ON RDS (SET 4 / 4)

The last set of experiments focus on LEAP's performance on RDS. Similarly, we present the evaluative metrics in Fig. 12.

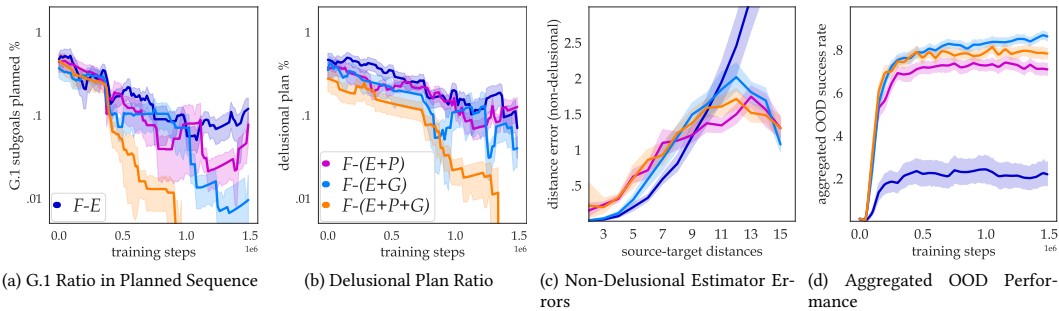

(a) G.1 Ratio in Planned Sequence    (b) Delusional Plan Ratio    (c) Non-Delusional Estimator Errors    (d) Aggregated OOD Performance

Figure 12: LEAP **on RDS**: each mean curve and CI (95%) bar are over 20 seed runs. **a)** Ratio of G.1 subgoals among the evolved sequence; **b)** Ratio of evolved sequences containing at least one G.1 target; **c)** The final estimation accuracies of non-delusional source-target pairs after training completed, across a spectrum of ground truth distances. In this figure, both distances (estimation and ground truth) are conditioned on the final version of the evolving policies; **d)** Each data point represents OOD evaluation performance aggregated over $4 \times 20$ newly generated tasks, with mean difficulty matching the training tasks.

The conclusions are similar, despite that the OOD performance gain by addressing delusions is significantly higher than in `SSM`.

### B.4.1 Breakdown of Task Performance

In Fig. 13, we present the evolution of LEAP variants' performance on the training tasks as well as the OOD evaluation tasks throughout the training process. Note that Fig. 12 **d)** is an aggregation of performances in Fig. 13 **b-e)**.

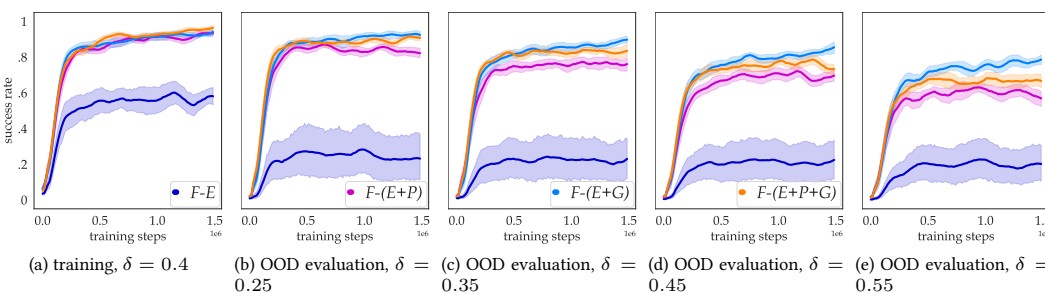

(a) training, $\delta = 0.4$
(b) OOD evaluation, $\delta = 0.25$
(c) OOD evaluation, $\delta = 0.35$
(d) OOD evaluation, $\delta = 0.45$
(e) OOD evaluation, $\delta = 0.55$

Figure 13: **Evolution of Performances of** LEAP **Variants on** `SSM`: each mean curve and CI (95%) bar are over 20 seed runs.

### B.5 Discussions & More Details of "generate" & "pertask"

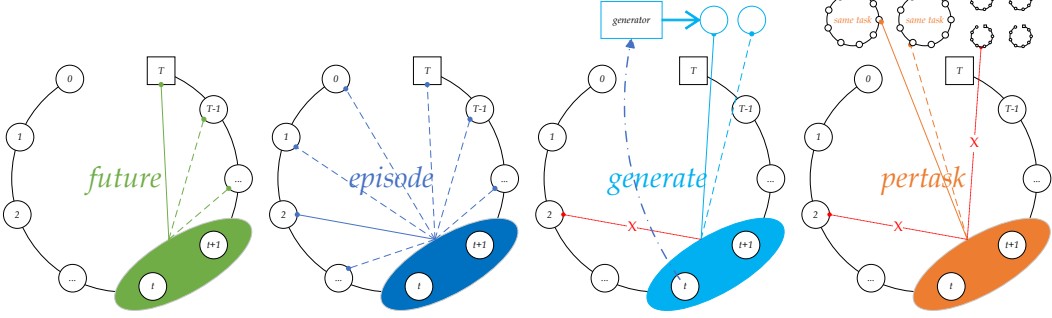

Figure 14: **Representative Atom Hindsight Relabeling Strategies & Newly Proposed Ones**: The first two strategies, "future" and "episode", are widely used as they create relabeled transitions that help estimators efficiently handle non-delusional planning. The last two, "generate" and "pertask", are effective at addressing delusions, making them useful in specific scenarios. Atomic hindsight strategies from the first group can serve as backbones for mixture strategies, complemented by the second group to address delusions.

### B.5.1 Implementation pf "pertask"

"pertask" takes the advantage for the fact that training is done on limited number of fixed task instances. We give each task a unique task identifier. At relabeling time, "pertask" samples observations among all the transitions marked with the same identifier as the current training task instance. This can be trivially implemented with individual auxiliary ERs that store only the experienced states with memory-efficient pointers to the buffered $x_t$'s in the main HER.

### B.5.2 Discussions

"generate" not only creates targets with G.1 targets, but also generate valid targets that should resemble the distribution it was trained on. Thus, it is not clear if mixing in data augmented by "generate" would result in lower sample efficiency in the estimation cases involving valid targets.

Take `SSM` as an example, "generate" seemed to have detrimental effect to non-delusional cases when applied to Skipper, while it greatly boosted accuracies for LEAP overall.

In some experiments, "pertask" demonstrated clear effectiveness in addressing E.1 as well, despite that it was not designed to. This is likely because of some generalization effects of the estimator, which were trained with additional data that boosted data diversity.

In some environments, we expect that "pertask" could also be used (for mixtures of the generator) to learn to generate longer-distance targets from the current states if the generator has trouble doing so with "future", with the accompanied risks of lower efficiency and G.2 hallucinations.

## C  IMPLEMENTATION DETAILS FOR EXPERIMENTS

### C.1  SKIPPER

Our adaptation of Skipper over the original implementation[1] in Zhao et al. (2024) is minimal. We have additionally added two simple vertex pruning procedures before the vertex pruning based on $k$-medoids. These two procedures include: 1) prune vertices that are duplicated, and 2) prune vertices that cannot be reached from the current state with the estimated connectivity.

We implemented a version of generator that can reliably handle both `RDS` and `SSM` with the same architecture. Please consult `models.py` in the submitted source code for its detailed architecture.

For `SSM` instances, since the state spaces are 4-times bigger than those of `RDS`, we ask that Skipper generate twice the number of candidates (both before and after pruning) for the proxy problems.

All other architectures and hyperparameters are identical to the original implementation.

For better adaptability during evaluation and faster training, Skipper variants in this paper keeps the constructed proxy problem for the whole episode during training and replanning only triggers a re-selection, while during evaluation, the proxy problems are always erased and re-constructed.

The quality of our adaptation of the original implementation can be assured by the fact the "F-E" variant's performance matches the original on `RDS`.

### C.2  LEAP

#### C.2.1  ADAPTATION FOR DISCRETE ACTION SPACES

LEAP's training involves two pretraining stages, that are, generator pretraining and distance estimator training.

We improved upon the adopted discrete-action space compatible implementation of LEAP (Nasiriany et al., 2019) from Zhao et al. (2024). We gave LEAP additional flexibility to use fewer subgoals along the way to the task goal if necessary. Also, we improved upon the Cross-Entropy Method (CEM), such that elite sequences would be kept intact in the next population during the optimization process. We increased the base population size of each generation to 512 and lengthened the number of iterations to 10.

For `RDS` $12 \times 12$ and `SSM` $8 \times 8$, at most 3 subgoals are used in each planned path. We find that employing more subgoals greatly increases the burden of CEM and lower the quality of the evolved subgoal sequences, leading to bad performance that cannot be effectively analyzed.

We used the same generator architecture and hyperparameters as in Skipper. All other architectures and hyperparameters remain unchanged.

Similarly for LEAP, for better adaptability during evaluation, the planned sequences of subgoals are always reconstructed whenever planning is triggered. While in training, the sequence is reused and only a subgoal selection is conducted.

The quality of our adaptation of the original implementation can be assured by the fact the "F-E" variant's performance matches the original on `RDS`.

---

[1] `https://github.com/mila-iqia/Skipper`

