# OpenReview forum: "Identifying and Addressing Delusions for Target-Directed Decision Making"
_ICLR.cc/2025/Conference — Submitted to ICLR 2025_

### Official Review · Reviewer_poEz · 2024-11-02

**Soundness:** 2
**Presentation:** 3
**Contribution:** 3
**Rating:** 6
**Confidence:** 3

**Summary:**

This paper tackles the challenge of delusions in target-directed reinforcement learning agents - a problem where agents develop false beliefs leading to unwanted behaviors and poor generalization. The authors identify and categorize different types of delusions (G.1, G.2, E.0, E.1, E.2), propose two new strategies ("generate" and "pertask") for addressing these delusions through hindsight relabeling, and introduce a hybrid approach that separates generator and estimator training. The work is validated through experiments on two environments (SSM and RDS) using two different methods (Skipper and LEAP), with detailed ablation studies and performance metrics.

**Strengths:**

* The paper's primary strengths lie in its clear problem identification, technical depth, and thorough empirical validation. The categorization of delusions is well-structured and the proposed solutions are practically implementable.
* The experimental work is comprehensive, with clear visualizations and detailed analysis. The writing is clear and well-organized, effectively using diagrams and examples to illustrate complex concepts.
* The hybrid approach of separating generator and estimator training demonstrates good understanding of the underlying challenges.

**Weaknesses:**

* The theoretical framework lacks formal analysis and guarantees about why the proposed strategies work. The experimental scope is confined to grid-world type of environments, missing opportunities to demonstrate effectiveness in more complex or real-world scenarios.
* The paper also lacks sufficient discussion of scalability challenges. How would these approaches scale to more complex environments with continuous action spaces? What theoretical guarantees can be provided about the reduction of delusions? How do these methods compare to other state-of-the-art approaches addressing similar issues in target-directed RL?

**Questions:**

Please address my questions in Weakness

---

> ### Author Response · Authors · 2024-11-15
> **Reply to Reviewer poEz (Part 1 / 3)**
>
> **_Thank you very much for the time and energy you have spent reviewing this paper._**
>
> **_To reply to you with deserving respect, we have organized your comments point-by-point so we will not leave ANY of your points behind and try our best to address all your concerns._**
>
> ----------
>
> `The theoretical framework lacks formal analysis and guarantees about why the proposed strategies work.`
>
> _We understand your concern. However, we feel that we can help justify the reasoning behind our choices which would hopefully lessen this concern. This is mostly because the strategies we proposed in this paper essentially act as **add-ons to existing methods**, and thus the guarantees and formal analysis should be grounded in each individual target-directed framework that our mitigation strategies (“generate” and “pertask”) are applied to and can only be done on an agent-by-agent basis. Note that our proposed strategies only provide training data counteracting delusions, the abilities to learn from them lie in individual agents._
>
> _In this work , we discussed extensively one necessary condition for learning against delusions (for general target-directed agents): having the correct update rules. And it is indeed the update rules that should provide guarantees for our mitigation strategies._
>
> _In target-directed frameworks, the update rules essentially learn about the relationship between the source state and the target state. This means that if certain update rules in certain target-directed framework have a convergence guarantee, then, with our mitigation strategies, the corresponding framework will be guaranteed to converge to the correct relationship in the source-target pair. The convergence should not be established without a detailed algorithm to be applied to._
>
> _This also means, without a convergence guarantee in the update rules, we will not be able to establish any other guarantees. This was why we did not include a theoretical analysis._
>
> _Take the method Skipper, used primarily for our experiments, for example. In the Skipper paper, the authors justified and proved that its learning rules would lead to a correct on-policy estimate of the expected distances between one state and another, i.e. the update rule regarding distance will learn how many timesteps the agent is expected to take, given its current policy, to reach a target state from a source state. In this case, if we use the proposed mitigation strategies “generate” or “pertask”, which guarantees the creation of source-G.1 and source-G.2 pairs if applicable, then the agent will be guaranteed to learn that from any valid state, it would take infinite timesteps to reach G.1 and G.2 targets, therefore converging delusional error to 0 (in terms of distances). Note that since the update rules in Skipper can only guarantee the convergence from a valid state to other states, not from an invalid state (G.1 or G.2) to others, we also carefully deconstructed the delusion errors into categories in the Appendix and validated the categorized convergences. A similar case can be made for the other method that we have tested as well, namely LEAP, which has its own update rules with its own convergence properties._
>
> _Following your point, we have added a condensed version of our arguments to the revised manuscript._
>
> _"Additionally, with a convergent learning rule (provided by the target-directed framework that these strategies are applied to), these strategies would also lead to the convergence to the correct estimation of G.1 and G.2 targets."_
>
> ***(TO BE CONTINUED)***

---

> ### Author Response · Authors · 2024-11-15
> **Reply to Reviewer poEz (Part 2 / 3)**
>
> `The experimental scope is confined to grid-world type of environments, missing opportunities to demonstrate effectiveness in more complex or real-world scenarios.`
>
> _We understand your concerns regarding the limited scope of experiments. We would like to highlight the following points that we feel can help explain/resolve this. 1) we are shackled by a chicken-and-egg problem in terms of finding other convincing experimental scenarios and present to you reasons 2) we are confident about the generality of our experiments and 3) why we are cautiously optimistic about the generality of our proposed strategies despite only using controlled experiments._
>
> **_Why we couldn’t present experiments in more complicated environments:_**
>
> _We chose not to include such experiments not because we didn’t try to do so, but rather because we tried with considerable effort yet could not find a convincing case (a good environment-agent combination) as of now, mostly because target-directed frameworks are under-researched and the appropriate evaluation scenarios are not quite accessible. Please consider the following points:_
>
> _First, we could not find a more complex or real-world-resembling environments that could give us the access to diagnosing delusional behaviors or could be compatible with OOD evaluation settings._
>
> _Second, we find that existing target-directed agents tend to fail completely if we introduce changes to their environments to check for their delusional behaviors. This is not to say that our proposed strategies make the algorithms fail, but that the existing agents have difficulty deploying to environments where delusions have consequences. For example, we tried to apply LEAP (originally implemented a very simple navigation task for Mujoco, simulated continuous-action control) on a changed Mujoco environment, where delusional behaviors are met with consequences (while the original environment it was tuned for was safe for the agent to have any wild targets because of the walls), the agent failed s._
>
> _We are in an academic environment where computational resources are very limited, thus by the time we submitted the work for review, we did not have other convincing results that would contribute to the narrative of this paper. We have not stopped working on this and hope that you understand such demand’s difficult nature._
>
> _The difficulty of presenting more experimental results was caused by the chicken-and-egg problem, which refers to the fact that target-directed agents are still novel to the field and are under-researched while problems like delusions hinder the research into this direction. We believe that the contribution of this work can be used to break the performance curses of target-directed agents, s.t. better performance will incentivize more research interests in this direction._
>
> **_Why we are confident about the existing experiments:_**
>
> _The complexity of our tasks is more than they seem to the eye. To understand this, we need to consider that the complexities of our tasks are focused on the reasoning challenges instead of learning from complex visual observations, as the never-seen-before layouts in each OOD evaluation task mount a combinatorial challenge to agents’ OOD generalization abilities. The truth is, the environments that we have employed and specifically in our OOD-oriented setting (training on few limited, testing on whole distribution) are very difficult for agents to solve. This was shown in the Skipper paper, where state-of-the-art methods like Director as well as competent baseline algorithms all suffer greatly on some grid-world tasks that are much simpler than the ones we used. In these environments, the agents cannot rely on memorization to solve evaluation tasks, instead can only use their understanding of the nature of the tasks._
>
> _With the difficult nature of our used tasks, we provided metrical examinations regarding delusional errors and behavior frequencies with rigor, acknowledged by **Reviewer 2JsJ**, demonstrating the significant effectiveness of our approach._
>
> **_Why we are cautiously optimistic about the generality of our proposed strategies:_**
>
> _Our main contributions focus on creating source-target pairs, where the targets can include both G.1. and G.2 targets, such that the agents could know how to deal with the problematic targets at decision-time when they are inevitably generated. The approaches we introduce do not rely on additional assumptions and can indeed guarantee the reduction of delusions when combined with a proper and convergent learning rule (guaranteed by the target-directed framework that the strategies are being applied to)._
>
> ***(TO BE CONTINUED)***

---

> > ### Author Response · Authors · 2024-11-15
> > **Reply to Reviewer poEz (Part 3 / 3)**
> >
> > _Take the method Skipper, used primarily for our experiments, for example. In the Skipper paper, the authors justified and proved that its learning rules would lead to a correct on-policy estimate of the expected distances between one state and another, i.e. the update rule regarding distance will learn how many timesteps the agent is expected to take, given its current policy, to reach a target state from a source state. In this case, if we use the proposed mitigation strategies “generate” or “pertask”, which guarantees the creation of source-G.1 and source-G.2 pairs if applicable, then the agent will be guaranteed to learn that from any valid state, it would take infinite timesteps to reach G.1 and G.2 targets, therefore converging delusional error to 0 (in terms of distances). Note that since the update rules in Skipper can only guarantee the convergence from a valid state to other states, not from an invalid state (G.1 or G.2) to others, we also carefully deconstructed the delusion errors into categories in the Appendix and validated the categorized convergences. A similar case can be made for the other method that we have tested as well, namely LEAP, which has its own update rules with its own convergence properties._
> >
> > _We believe that our contribution’s flexibility and lack of additional assumptions indicate its generality in target-directed frameworks, and we have not found even one convincing counterexample to the applicability of our strategies. On this note, we remain cautiously optimistic, despite that we were shackled by a chicken-and-egg problem in terms of experiments._
> >
> >
> > `The paper also lacks sufficient discussion of scalability challenges. How would these approaches scale to more complex environments with continuous action spaces? What theoretical guarantees can be provided about the reduction of delusions? How do these methods compare to other state-of-the-art approaches addressing similar issues in target-directed RL?`
> >
> > **_On how would these approaches scale to more complex environments with continuous action spaces:_**
> >
> > _Our approaches do not assume discrete action spaces nor environment complexities. As long as the target-directed agents our mitigation strategies are applied to are compatible with the environment, we do not see any reason why they would cease to work. We added a clarification statement to the revised manuscript:_
> >
> > _"Note that since our mitigation strategies do not introduce additional assumptions, they should be expected to be applicable generally."_
> >
> > **_On what theoretical guarantees can be provided about the reduction of delusions:_**
> >
> > _As we have explained in the previous reply, the theoretical guarantees lie in the individual methods. If a method has guarantees to converge to correct estimates, then, since our proposed mitigation strategies are guaranteed to provide source-G.1 and source-G.2 pairs if applicable, the corresponding method will converge to address delusions. We added these statements explicitly to our revised manuscript._
> >
> > **_On how do these methods compare to other state-of-the-art approaches addressing similar issues in target-directed RL?_**
> >
> > _To the best of our knowledge, there are no other existing methods in addressing delusions in target-directed RL. There are very limited conceptual discussions about how to address hallucinated targets in the intersections of AI safety and RL. Ours is the first paper to abstract the target-directed framework to understand why these agents are prone to chasing hallucinated targets and proposed strategies to mitigate the issue. In existing literature, such as LEAP, there are workaround engineering techniques proposed to create more diverse source-target pairs in the training data to improve performance generally. However, the existing literature never identified the root problems nor addressed them systematically._
> >
> > _We have talked about these extensively in the related works section._
> >
> > -------
> >
> > **_We have done our best to address all your concerns thoroughly and, we hope, to your satisfaction. Our sincere intent in completing this research is to bring greater awareness of delusions in agent behavior to the research community. We hope this work will help reduce the time and energy spent on recurring issues, such as creating delusional agents and misunderstanding their failures, thereby accelerating research progress and enhancing the impact of our field._**
> >
> > **_We sincerely wish that you could consider raising the review score, since we believe that you approve the contributions of this work, and the score is barely below the targeted acceptance threshold._**
> >
> > **_Thank you very much for your consideration!_**

---

> ### Author Response · Authors · 2024-12-04
> **Reminder for Discussion Period**
>
> Dear Reviewer,
>
> We hope you are doing well!
> We put effort into addressing your comments and hope you could take some time to participate in the discussion period, as the period is about to end soon.
>
> Best,
>
> Authors

---

### Official Review · Reviewer_fmgU · 2024-11-03

**Soundness:** 2
**Presentation:** 2
**Contribution:** 2
**Rating:** 5
**Confidence:** 1

**Summary:**

The paper addresses delusional behaviors in reinforcement learning (RL) agents that set their own sub-goals (targets). These agents sometimes form "delusions"—incorrect beliefs about which targets are achievable or safe. The study identifies two types of target-related delusions: G.1 (nonexistent or invalid) and G.2 (temporarily unreachable). It also categorizes types of estimator errors that arise from delusional target evaluations: E.0 (non-delusional target errors), E.1 (errors related to G.1 targets), and E.2 (errors related to G.2 targets). To combat these, the paper introduces modifications to Hindsight Experience Replay (HER) to train agents to recognize problematic targets more effectively. This approach demonstrates some improvement in out-of-distribution (OOD) generalization, though specific results vary.

**Strengths:**

Novelty of the Problem: The paper tackles a unique issue in reinforcement learning—delusions in target-directed decision-making, which hasn't been addressed explicitly in previous literature.

Detailed Taxonomy of Delusions: The categorization into G.1 and G.2 target types, along with E.0, E.1, and E.2 estimator delusions, is well-structured and provides clear, actionable insights into RL challenges.

**Weaknesses:**

Complexity and Clarity: The paper introduces complex terminology and presents dense explanations that may be challenging to follow. This affects readability and accessibility, as it requires a high level of familiarity with RL and HER.

Limited Generalization Evidence: Although the study introduces methods to handle delusional targets, the empirical validation is largely confined to specific, controlled environments (SSM and RDS). It is unclear if these strategies would generalize well to more complex or real-world scenarios.

High Computational Cost: The proposed strategies, particularly the "generate" strategy, rely on additional computational resources. This limitation may hinder real-world application due to high resource requirements.

**Questions:**

I am not an expert in this domain, and I found this paper relatively difficult to follow. It seems to be more of a conceptual paper rather than a traditional ML paper with detailed formulas and technical explanations. Given this, I don’t have any further questions.

---

> ### Author Response · Authors · 2024-11-15
> **Reply to Reviewer fmgU (Part 1 / 2)**
>
> **_Thank you very much for the time and energy you have spent reviewing this paper._**
>
> **_To reply to you with deserving respect, we have organized your comments point-by-point so we will not leave ANY of your points behind and try our best to address all your concerns._**
>
> ----------
>
> `Complexity and Clarity: The paper introduces complex terminology and presents dense explanations that may be challenging to follow. This affects readability and accessibility, as it requires a high level of familiarity with RL and HER.`
>
> _We apologize for the challenges for following through on the paper material. Following your suggestions, we have edited the manuscript very HEAVILY, s.t. an intuitive and clear picture can be painted for your better accessibility. **Please consider reading parts of the revised manuscript to see if our changes addressed your concerns**. We prioritized on reducing complex terminologies and reducing cognitive load, for example in the preliminary section, etc._
>
> `Limited Generalization Evidence: Although the study introduces methods to handle delusional targets, the empirical validation is largely confined to specific, controlled environments (SSM and RDS). It is unclear if these strategies would generalize well to more complex or real-world scenarios.`
>
> _We understand your concerns regarding the limited scope of experiments. We would like to argue the following points in more detail. 1) we are shackled by a chicken-and-egg problem in terms of finding other convincing experimental scenarios and present to you reasons 2) the reasons why we are confident about the generality of our experiments and 3) why we are cautiously optimistic about the generality of our proposed strategies despite only using controlled experiments._
>
> **_Why we couldn’t present experiments more complicated environments:_**
>
> _We chose not to present such experiments not because we didn’t try to do so, instead, we tried with considerable effort yet could not find a convincing case (a good environment-agent combination) as of now, mostly because target-directed frameworks are under-researched and the appropriate evaluation scenarios are not quite accessible. Please consider the following points:_
>
> _First, we could not find a more complex or real-world-resembling environment that could give us the access to diagnosing delusional behaviors or could be compatible with OOD evaluation settings._
>
> _Second, we find that existing target-directed agents tend to fail completely if we introduce changes to their environments to check for their delusional behaviors. This is not to say that our proposed strategies make the algorithms fail, but that the existing agents have difficulty deploying to environments where delusions have consequences. For example, we tried to apply LEAP (originally implemented a very simple navigation task for Mujoco, simulated continuous-action control) on a changed Mujoco environment, where delusional behaviors are met with consequences (while the original environment it was tuned for was safe for the agent to have any wild targets because of the walls), the agent failed miserably._
>
> _We are in an academic environment where computational resources are very limited, thus by the time we submitted the work for review, we did not have other convincing results that would contribute to the narrative of this paper. We have not stopped working on this and hope that you understand such demand’s difficult nature._
>
> _The difficulty of presenting more experimental results was caused by the chicken-and-egg problem, which refers to the fact that target-directed agents are still novel to the field and are under-researched, while problems like delusions hinder the research into this direction. We believe that the contribution of this work can be used to break the performance curses of target-directed agents, s.t. better performance will incentivize more research interests in this direction._
>
> **_Why we are confident about the existing experiments:_**
>
> ***(TO BE CONTINUED)***

---

> > ### Author Response · Authors · 2024-11-15
> > **Reply to Reviewer fmgU (Part 2 / 2)**
> >
> > _The complexity of our tasks is more than they seem to the eye. To understand this, we need to consider that the complexities of our tasks are focused on the reasoning challenges instead of learning from complex visual observations, as the never-seen-before layouts in each OOD evaluation task mount a combinatorial challenge to agents’ OOD generalization abilities. The truth is, the environments that we have employed and specifically in our OOD-oriented setting (training on few limited, testing on whole distribution) are very difficult for agents to solve. This was shown in the Skipper paper, where state-of-the-art methods like Director as well as competent baseline algorithms all suffer greatly on some grid-world tasks that are much simpler than the ones we used. In these environments, the agents cannot rely on memorization to solve evaluation tasks, and instead can only use their understanding of the nature of the tasks._
> >
> > _With the difficult nature of our used tasks, we provided metrical examinations regarding delusional errors and behavior frequencies with rigor, acknowledged by **Reviewer 2JsJ**, demonstrating the significant effectiveness of our approach._
> >
> > **_Why we are cautiously optimistic about the generality of our proposed strategies:_**
> >
> > _Our main contributions focus on creating source-target pairs, where the targets can include both G.1. and G.2 targets, such that the agents could know how to deal with the problematic targets at decision-time when they are inevitably generated. The approaches we introduce do not rely on additional assumptions and can indeed guarantee the reduction of delusions when combined with a proper and convergent learning rule (guaranteed by the target-directed framework that the strategies are being applied to)._
> >
> > _Take the method Skipper, used primarily for our experiments, for example. In the Skipper paper, the authors justified and proved that its learning rules would lead to a correct on-policy estimate of the expected distances between one state and another, i.e. the update rule regarding distance will learn how many timesteps the agent is expected to take, given its current policy, to reach a target state from a source state. In this case, if we use the proposed mitigation strategies “generate” or “pertask”, which guarantees the creation of source-G.1 and source-G.2 pairs if applicable, then the agent will be guaranteed to learn that from any valid state, it would take infinite timesteps to reach G.1 and G.2 targets, therefore converging delusional error to 0 (in terms of distances). Note that since the update rules in Skipper can only guarantee the convergence from a valid state to other states, not from an invalid state (G.1 or G.2) to others, we also carefully deconstructed the delusion errors into categories in the Appendix and validated the categorized convergences. A similar case can be made for the other method that we have tested as well, namely LEAP, which has its own update rules with its own convergence properties._
> >
> > _We believe that our contribution’s flexibility and lack of additional assumptions indicate its generality in target-directed frameworks, and we have not found even one convincing counterexample to the applicability of our strategies. On this note, we remain cautiously optimistic, despite the fact that we were shackled by a chicken-and-egg problem in terms of experiments._
> >
> > `High Computational Cost: The proposed strategies, particularly the "generate" strategy, rely on additional computational resources. This limitation may hinder real-world application due to high resource requirements.`
> >
> > _We acknowledge this point, and added emphasis explicitly to this point to the end of the subsection that introduces “generate”:_
> >
> > _"'generate' comes at a cost of extra computation, to which attention may be required in real-world applications with speed demands."_
> >
> > _Previously, this was only stated in Table 1._
> >
> > -------
> >
> > **_We understand that the manuscript was challenging for you to read. Thus, we spent the past few days to do our best to rewrite it, hopefully to your satisfaction. We hope that you could consider raising the review score, since we believe that you approve the contributions of this work, and the score is barely below the targeted acceptance threshold._**
> >
> > **_With all honesty, we completed this work with the hope that it could help to bring awareness of delusions to our research community so all of us could reduce time and energy wasted on repeating these mistakes. We believe that this contribution deserves more consideration because of its positive impact on our understanding of computational decision-making._**
> >
> > **_Thank you very much for your consideration!_**

---

> ### Author Response · Authors · 2024-12-04
> **Reminder for Discussion Period**
>
> Dear Reviewer,
>
>
> We hope you are doing well!
> We put effort into addressing your comments and hope you could take some time to participate in the discussion period, as the period is about to end soon.
>
> Best,
>
> Authors

---

### Official Review · Reviewer_XvdY · 2024-11-03

**Soundness:** 1
**Presentation:** 2
**Contribution:** 2
**Rating:** 5
**Confidence:** 3

**Summary:**

This paper addresses failure modes in target-directed Reinforcement Learning (RL) agents induced by delusions, where the agent holds false beliefs about their targets, leading to poor OOD generalization and/or undesired behaviours.
The authors first identify and characterize delusions that either come from the generator or the estimator of the targets. The empirical experiments are based in the case of HER-based training.
In this case, they propose 2 new strategies: 'generate', 'pertask' in addition to 2 ones  coming from the literature ('future', 'episode').
The authors combine in an hybrid fashion this 4 HER strategies, showing that this approach can help mitigate the issues coming from delusions.

**Strengths:**

- OOD generalization is an important topic in RL, and further work on this is beneficial for the community.
- The characterization of different type of delusions (coming either from the generator or estimator) is valuable both in the context of the paper, but also for future work possibly building on this.
- The approach presented in Section 4 is sounds, Table 1 is particularly useful to understand the motivation behind their design choices.

**Weaknesses:**

In general, I am particularly unsure about both the relevance and the soundness of the claim. The paper in fact argues very strongly about their approach, which I however think is limited in some ways. In fact, the authors investigate only HER-based training, specifically the case where both the estimator and the generator are trained in this way. It is okay if the paper focuses only on this, but the authors claims that all of their approaches can be extended beyond HER, which is not easily validated in my opinion.

The paper is an empirical paper, although it does not seem the Experiment section to be sufficiently strong. The paper does not explain the results extensively and/or in enough detail everywhere. I'm also skeptical about the results being as strong as they claimed they are.  In general I find the figures hard to follow.

Please see below for more details for my criticisms about the paper.

**Questions:**

- "For aggregated OOD performance [...] we sample 20 tasks [...], which have a mean difficulty matching the training tasks.". Could you clarify why you call this OOD performance? For this to be the case, I would both expect the gradient of difficulties to be wider. My intuition is that you want to keep the average the same to distinguish capability generalization vs delusion generalization, is that correct? I would however not call this OOD performance if I understand it correctly...  In general, I am curious how you see the difference between capability vs goal misgeneralization [1], and argue which of the two your work addresses. This would help me better clarify how valid the claims made are

- "To maximize difficulty, the initial state is fixed in each novel evaluation task instance: the agents are not only spawned to be at the furthest side of the monster": this is not always true, especially if the agent did not see often particular combination of start positions/state. This is true in the case only in terms of capability generalization, but I guess your answer to the above can also address this one.

- I suggest for the experiments to include actual OOD scenarios as I intend them. For example, try to test a scenario where the agent spawns with already one shield in hand, but another is present in the environment ( without training in such a situation). It would then be interesting to see if the targets generated still suggest to pick up the other shield even if it is in fact not necessary...

- I'm unsure the characterization of delusions proposed is comprehensive - could you argue why this is the case? It seems that this applies only to the environments proposed, and may not be general. It is okay if this is the case, but that should be specified better.

- I would like to see an experiment where the proposed approaches are validated beyond HER. If that is not possible, I argue to be appropriate to weaken the claims regarding such of a possible extension. If you do not agree, could you please explain why we expect these results to hold further?

- Figure 3 is too condensed, and very hard too look at. In some cases the error bars are extremely large, which makes difficult to even consider some of the results statistically valid. I suggest this work [2] to improve how metrics are reported such that to have a better sense if this empirical results do in fact hold...

- The best performing algorithm in Figure 3h seem to reach accuracy ~0.4. If I understand correctly, the maximum achievable in that case is 1, correct? If you have discounting, what is the baseline given by the optimal policy? Are all levels proposed solvable? It is important to report a baseline also for the other figures somehow, since the results presented in this way make it difficult for their significance to be judged.

- I suggest moving the related work section either after the introduction, or before the conclusion. Now it breaks the flow of the paper since it is placed just before the experiments

- The authors say in line 371 "Due to page limit, 3 out of 4 sets of experiments are only presented in the Appendix". While I do not expect all of the experiments to fit in the main paper, being able to condense and critically decide what goes in the main text is an important task authors should dedicate time to. This is mainly an empirical paper, and it is thus important for the most relevant experiments to be in the main text such that the claims can be supported by those experiments.

- ' generators are a major source of risk related to delusion' should at least backed up by some previous work.

There are also many parts with grammatical errors, e.g.
- line 186
-line 251
-line 257 etc..




[1] Di Langosco, Lauro Langosco, et al. "Goal misgeneralization in deep reinforcement learning." International Conference on Machine Learning. PMLR, 2022.

[2] Agarwal, Rishabh, et al. "Deep reinforcement learning at the edge of the statistical precipice." Advances in neural information processing systems 34 (2021): 29304-29320.

---

> ### Author Response · Authors · 2024-11-15
> **Reply to Reviewer XvdY (Part 1 / 6)**
>
> **_Thank you very much for the time and energy you have spent reviewing this paper._**
>
> **_To reply to you with deserving respect, we have organized your comments point-by-point so we will not leave ANY of your points behind and try our best to address all your concerns._**
>
> ----------
>
> `In general, I am particularly unsure about both the relevance and the soundness of the claim. The paper in fact argues very strongly about their approach, which I however think is limited in some ways. In fact, the authors investigate only HER-based training, specifically the case where both the estimator and the generator are trained in this way. It is okay if the paper focuses only on this, but the authors claims that all of their approaches can be extended beyond HER, which is not easily validated in my opinion.`
>
> _We understand your concerns and would like to provide you with our perspective of why we insisted on the existing forms of our claims. We have to be first on the same page by recognizing that: our claim was not that our strategies could be applied to all methods, but rather only to target-directed methods, which are all governed by the key notion of source-target pairs._
>
> **_The Foundation of Our Approach is on Source-Target pairs, but not HER_**
>
> _Our proposed mitigation strategies revolve around source-target pairs, which can be created and easily explained by HER. However, our strategies do not rely on the mechanisms of HER. Even though in literature HER was mainly viewed from the perspective of a sample-efficiency enhancing technique, it is fundamentally_ _a minimalist method of creating source-target pairs_ _that enable the agent to learn the relationship between the source state and the target state, as we have discussed in the manuscript. Such a relationship (the idea of source-target pair) is universal in target-directed frameworks thus the reason of our claim._
>
> _We understand that you have concerns that HER training would be representative and an indication of all potential training methods. However, we chose HER not only because of its popularity in target-directed agents, but also, most importantly due to its direct correspondence with the source-target pairs, fundamental to target-directed agents._
>
> _We understand that this point may not have been sufficiently clear when reading the manuscript, thus we have spent the past few days trying to improve our writing regarding this point. For example, we now have the following updated statements in the revised manuscript:_
>
> _"Essentially, target-directed agents are trained on 'source-target pairs', which are organized training data that could let the agents learn about the relationship between the source state and the target state."_
>
> _We thank you for prompting this modification through your comments._
>
> **_HER relabeling can be transformed trivially into general training losses_**
>
> _An important fact is that any HER relabeling mechanism can be implemented in the form of an auxiliary loss established over two sampled transitions from an ordinary experience replay. This not only shows a direct equivalence between relabeling and additional losses but also shows that the relabeling strategies we have proposed can be trivially transformed and incorporated into existing agents. Combined with the fact that our approaches did not introduce any additional assumptions, these strongly indicate the generality of our approach, as our strategies can be directly translated to non-HER auxiliary losses applicable to general agents._
>
> _We understand that this point could be difficult to understand, thus we have added more statements in the revised manuscript for a clearer presentation of our contributions. For example:_
>
> _"Having identified delusions, taking advantage of the shared reliance on source-target pairs, we can develop strategies applicable to general target-directed agents coming from various training procedures. Then, we provide examples on how to materialize such strategies in agents trained with HER, because of its popularity, its direct association with source-target pairs and the fact that one can trivially transform HER-based training into losses established over two sampled transitions from ordinary experience replays."_
>
> **_It does not matter if both the generator and the estimator were trained with HER_**
>
> _In this work, we abstracted an agent that utilizes HER to train both the generator and the estimator, only because we could use it to deliver clearer and more unified arguments._
>
> _After the explanation we have given above, it should be clear that the mitigation strategies do not rely on HERs. For example, even if a generator is trained with autoregressive or unconditional losses, as long as it fits into the target-directed framework, we can apply our strategies onto the estimators to deliver the results._
>
> ***(TO BE CONTINUED)***

---

> > ### Author Response · Authors · 2024-11-15
> > **Reply to Reviewer XvdY (Part 2 / 6)**
> >
> > _Training generator with “future”-HER essentially means training it to generate targets corresponding to future states, which can be done in a variety of different ways, such as autoregressively. Meanwhile, training generators with “episode”-HER means asking the generators to additionally re-create targets corresponding to past states, which can also be done in many ways, for example, training an unconditional VAE, as in LEAP._ Our experiments clearly demonstrate that our approaches are applicable to all these scenarios.
> >
> > _We heavily edited the manuscript to make the points above clearer, for example, in the preliminary section we now have: "_ _Note that even for methods whose generators are not trained with HER, our estimator-focused strategies still apply."_
> >
> > `The paper is an empirical paper, although it does not seem the Experiment section to be sufficiently strong. The paper does not explain the results extensively and/or in enough detail everywhere. I'm also skeptical about the results being as strong as they claimed they are. In general I find the figures hard to follow.`
> >
> >
> > _We understand your concerns regarding the limited scope of experiments. We want to explain to you that 1) we are shackled by a chicken-and-egg problem in terms of finding other convincing experimental scenarios and present to you reasons 2) why we are confident about the generality of our experiments and 3) why we are cautiously optimistic about the generality of our proposed strategies despite only using controlled experiments._
> >
> > **_Why we couldn’t present experiments more complicated environments:_**
> >
> > _We chose not to not because we didn’t try to do so, instead, we tried with considerable effort yet could not find a convincing case (a good environment-agent combination) as of now, mostly because target-directed frameworks are under-researched and the appropriate evaluation scenarios are not quite accessible. Please consider the following points:_
> >
> > _First, we could not find a more complex or real-world-resembling environments that could give us the access to diagnosing delusional behaviors or could be compatible with OOD evaluation settings._
> >
> > _Second, we find that existing target-directed agents tend to fail completely if we introduce changes to their environments to check for their delusional behaviors. This is not to say that our proposed strategies make the algorithms fail, but that the existing agents have difficulty deploying to environments where delusions have consequences. For example, we tried to apply LEAP (originally implemented a very simple navigation task for Mujoco, simulated continuous-action control) on a changed Mujoco environment, where delusional behaviors are met with consequences (while the original environment it was tuned for was safe for the agent to have any wild targets because of the walls), the agent failed miserably._
> >
> > _We are in an academic environment where computational resources are very limited, thus by the time we submitted the work for review, we did not have other convincing results that would contribute to the narrative of this paper. We have not stopped working on this and hope that you understand such demand’s difficult nature._
> >
> > _The difficulty of presenting more experimental results was caused by the chicken-and-egg problem, which refers to the fact that target-directed agents are still novel to the field and are under-researched while problems like delusions hinder the research into this direction. We believe that the contribution of this work can be used to break the performance curses of target-directed agents, s.t. better performance will incentivize more research interests in this direction._
> >
> > **_Why we are confident about the existing experiments:_**
> >
> > _The complexity of our tasks is more than they seem to the eye. To understand this, we need to consider that the complexities of our tasks are focused on the reasoning challenges instead of learning from complex visual observations, as the never-seen-before layouts in each OOD evaluation task mount a combinatorial challenge to agents’ OOD generalization abilities. The truth is, the environments that we have employed and specifically in our OOD-oriented setting (training on few limited, testing on whole distribution) are very difficult for agents to solve. This was shown in the Skipper paper, where state-of-the-art methods like Director as well as competent baseline algorithms all suffer greatly on some grid-world tasks that are much simpler than the ones we used. In these environments, the agents cannot rely on memorization to solve evaluation tasks, instead can only use their understanding of the nature of the tasks._
> >
> > ***(TO BE CONTINUED)***

---

> > > ### Author Response · Authors · 2024-11-15
> > > **Reply to Reviewer XvdY (Part 3 / 6)**
> > >
> > > _With the difficult nature of our used tasks, we provided metrical examinations regarding delusional errors and behavior frequencies with rigor, acknowledged by **Reviewer 2JsJ**, demonstrating the significant effectiveness of our approach._
> > >
> > > **_Why we are cautiously optimistic about the generality of our proposed strategies:_**
> > >
> > > _Our main contributions focus on creating source-target pairs, where the targets can include both G.1. and G.2 targets, such that the agents could know how to deal with the problematic targets at decision-time when they are inevitably generated. The approaches we introduce do not rely on additional assumptions and can indeed guarantee the reduction of delusions when combined with a proper and convergent learning rule (guaranteed by the target-directed framework that the strategies are being applied to)._
> > >
> > > _Take the method Skipper, used primarily for our experiments, for example. In the Skipper paper, the authors justified and proved that its learning rules would lead to a correct on-policy estimate of the expected distances between one state and another, i.e. the update rule regarding distance will learn how many timesteps the agent is expected to take, given its current policy, to reach a target state from a source state. In this case, if we use the proposed mitigation strategies “generate” or “pertask”, which guarantees the creation of source-G.1 and source-G.2 pairs if applicable, then the agent will be guaranteed to learn that from any valid state, it would take infinite timesteps to reach G.1 and G.2 targets, therefore converging delusional error to 0 (in terms of distances). Note that since the update rules in Skipper can only guarantee the convergence from a valid state to other states, not from an invalid state (G.1 or G.2) to others, we also carefully deconstructed the delusion errors into categories in the Appendix and validated the categorized convergences. A similar case can be made for the other method that we have tested as well, namely LEAP, which has its own update rules with its own convergence properties._
> > >
> > > _We believe that our contribution’s flexibility and lack of additional assumptions indicate its generality in target-directed frameworks, and we have not found even one convincing counterexample to the applicability of our strategies. On this note, we remain cautiously optimistic, despite that we were shackled by a chicken-and-egg problem in terms of experiments._
> > >
> > > `"For aggregated OOD performance [...] we sample 20 tasks [...], which have a mean difficulty matching the training tasks.". Could you clarify why you call this OOD performance? For this to be the case, I would both expect the gradient of difficulties to be wider. My intuition is that you want to keep the average the same to distinguish capability generalization vs delusion generalization, is that correct? I would however not call this OOD performance if I understand it correctly ... In general, I am curious how you see the difference between capability vs goal misgeneralization [1], and argue which of the two your work addresses. This would help me better clarify how valid the claims made are`
> > >
> > > _The OOD setting we adopted in our work is only a subset of OOD generalization challenges, which we have motivated at the beginning of the paper. Very importantly, we want to emphasize that this setting has been long-accepted and widely adopted in literature._
> > >
> > > _We adopted the OOD evaluation criteria popularized by ProcGen (the non-curriculum setting), which was also later widely used including in the Skipper framework, which our experiments were built upon. The idea is that an agent is trained in a limited few tasks while being evaluated on a whole distribution of tasks with the same nature and it seeks to question if an agent could learn a truly generalizable skill, applicable to even previously unseen scenarios of the same nature. This same-nature-zero-shot OOD evaluation corresponds directly to the challenge of distributional shifts and discrepancies between the training and evaluation tasks motivated at the beginning of the paper._
> > >
> > > _In the capability misgeneralization v.s. goal misgenearlization failure categorization, the OOD generalization failure is attributed into the two factors, corresponding to the competence of implementing targets v.s. the correctness of targets, in the language of this work. While such is a sensible abstraction from a behavioral standpoint, our work was not established on this abstraction. However, we are happy to discuss with you the connections:_
> > >
> > > _Reducing hallucination directly corresponds to reducing goal misgeneralization. However, in target-directed frameworks, goal misgeneralization is also reduced by the estimator that can reject “misgeneralized targets”._
> > >
> > > ***(TO BE CONTINUED)***

---

> > > > ### Author Response · Authors · 2024-11-15
> > > > **Reply to Reviewer XvdY (Part 4 / 6)**
> > > >
> > > > _Reducing delusions about targets (with our proposed strategies) reduces both capability misgeneralization and goal misgeneralization. To see both sides, we must point out that 1) evaluating a proposed target requires capability generalization, since it includes the understanding of if the agent can reach certain target and how the agent would reach it (tied to the agent’s capabilities); and 2) less delusions about targets in target-directed frameworks means problematic targets are less likely favored._
> > > >
> > > > _We hope the above points answered your questions and you could understand that our OOD setting was adopted from approved peer literature and the connections between our viewpoints on generalization to the work you have mentioned._
> > > >
> > > > `"To maximize difficulty, the initial state is fixed in each novel evaluation task instance: the agents are not only spawned to be at the furthest side of the monster": this is not always true, especially if the agent did not see often particular combination of start positions/state. This is true in the case only in terms of capability generalization, but I guess your answer to the above can also address this one.`
> > > >
> > > > _We hope that the previous reply shows that our work was not built upon the capability v.s. goal misgeneralization dichotomy. In the quoted statement, we were simply saying that, instead of letting the agent only solve a part of the task, as in training, in OOD evaluation, the agents are asked to solve the previously unseen tasks completely, from the start to finish (instead of only finishing parts of it, as in the training tasks)._
> > > >
> > > > `I suggest for the experiments to include actual OOD scenarios as I intend them. For example, try to test a scenario where the agent spawns with already one shield in hand, but another is present in the environment (without training in such a situation). It would then be interesting to see if the targets generated still suggest to pick up the other shield even if it is in fact not necessary...`
> > > >
> > > > _We understand and respect your curiosity. However, our paper’s focus and claims are in fact well-aligned its own experimental settings. There are many flavors in OOD generalization, and we think that our adopted setting, a setting long existed in literature, should also deserve its merit._
> > > >
> > > > _Your suggested OOD scenario translates to an agent generalizing its learned skills in tasks NOT sharing the same nature with the training tasks, as the agents were never trained to deal with cases where two shields come into play. This means effectively the agent would be evaluated in a space of unseen G.1 targets. This is an ambitious idea that would require more consideration. Sincerely, we would be very curious to know in our future work._
> > > >
> > > > _We are open to investigate your intended OOD scenarios for future work, however, it is not practical for us to realign and do experiments in the short amount of time we were given in the discussion period. We hope that you can kindly consider this matter and could approve of our adopted settings, which served our purpose well, as suggested by **Reviewer 2JsJ**._
> > > >
> > > > `I'm unsure the characterization of delusions proposed is comprehensive - could you argue why this is the case? It seems that this applies only to the environments proposed, and may not be general. It is okay if this is the case, but that should be specified better.`
> > > >
> > > > _While we acknowledge that our categorization may not be exhaustive in the SUB-categories, we would present our reasons on why the overall categorization is justifiably comprehensive._
> > > >
> > > > _Formally, when a categorization is governed by a categorical indicator, “partitioning” occurs, leading to a series of disjoint sets whose union is the full set, i.e. leading to a comprehensive categorization. The indicators we used were “reachability” and “temporal-relations”, i.e. if a target is reachable, ALWAYS unreachable or temporarily unreachable. There are no other omitted categories. Specifically:_
> > > >
> > > > _First, we divided targets into achievable ones and problematics ones. Within the problematic ones, we partitioned them by the temporal-validity of the targets into the two types: G.1 targets are the ones that can never be fulfilled while G.2 targets are the ones that cannot be fulfilled from a current state yet can be fulfilled from other states. The partition of problematic targets based on the temporal-relationship is exhaustive. In other words, a problematic target is either G.1. or G.2._
> > > >
> > > > _For the three types of targets, we identified the corresponding delusions E.0, E.1 and E.2, which are related to valid targets, G.1 targets and G.2 targets respectively._
> > > >
> > > > _Based on these partitions, we cannot see a basis why our categorization can be incomplete._
> > > >
> > > > ***(TO BE CONTINUED)***

---

> > > > > ### Author Response · Authors · 2024-11-15
> > > > > **Reply to Reviewer XvdY (Part 5 / 6)**
> > > > >
> > > > > _We should also point out that, in the conclusion section, we have already acknowledged that within each type, e.g. G.1, there could be sub-categories with distinct characteristics. We are open to receiving counterarguments or counter-examples and remain cautiously confident about our claims._
> > > > >
> > > > > _We have added more descriptions to clarify the categorization in the revised manuscript and hopefully the readers would not be confused about our statements._
> > > > >
> > > > > `I would like to see an experiment where the proposed approaches are validated beyond HER. If that is not possible, I argue to be appropriate to weaken the claims regarding such of a possible extension. If you do not agree, could you please explain why we expect these results to hold further?`
> > > > >
> > > > > _We have answered the points in the previous reply. We hope that this answered your concerns. We would be happy to do such an experiment in the future work, as we think that we have already quite convincing results in a dominantly popular method with profound generality._
> > > > >
> > > > > _Notably, the idea behind “generate” was initially proposed as an auxiliary loss applicable to generic target-directed agents in one of the cited papers. Because of the aforementioned direct equivalence between two-sample loss and HER, we successfully translated “generate” into an HER strategy._
> > > > >
> > > > > `Figure 3 is too condensed, and very hard too look at. In some cases the error bars are extremely large, which makes difficult to even consider some of the results statistically valid. I suggest this work [2] to improve how metrics are reported such that to have a better sense if this empirical results do in fact hold...`
> > > > >
> > > > > _We acknowledge this issue and would like to provide you with more details to see that we were cautious about our claims._
> > > > >
> > > > > _We in fact used the same reporting method as in Fig.10 of [2] (mean curves with 95% CI). The fact that some cases, mostly Fig.3 c), have large error bars was because G.1-E.1 behaviors are rare to begin with (delusional errors do not lead to delusional behaviors all the time) and the variances were indeed high to begin with._
> > > > >
> > > > > _We didn’t have more computational resources to conduct more than 20 seed runs for each variant (we are in an academic setting without too much computational resources)._
> > > > >
> > > > > _We were also very careful about our statements. We did not use “significant” when describing the reduction in delusional behavior frequencies precisely because of the overlap of the confidence intervals. We only used the word “significant” when describing the statistically significant (alpha=0.05) pairwise improvements in the delusional errors in estimation and the OOD performances._
> > > > >
> > > > > _The overlapping error bars are mostly inter-group and not intra –group (the baseline group v.s. the variants group). This means that the variants such as F-(E+P) generally have little overlap, especially as training progresses, with baselines such as F-E. For clear comparison, we intended that the readers observe the curves in pairs, one from the baseline group and another from the variant group._
> > > > >
> > > > > _Following your suggestions and similar ones from the other reviewers, we have tried our best to improve on the visual presentations, and we have double-checked the accuracy of our results, in accordance with [2]._
> > > > >
> > > > > `The best performing algorithm in Figure 3h seem to reach accuracy ~0.4. If I understand correctly, the maximum achievable in that case is 1, correct? If you have discounting, what is the baseline given by the optimal policy? Are all levels proposed solvable? It is important to report a baseline also for the other figures somehow, since the results presented in this way make it difficult for their significance to be judged.`
> > > > >
> > > > > _We understand your concern and would like to clarify some misunderstandings._
> > > > >
> > > > > _The performance was about success rate, therefore not discounted, and the tasks are all solvable. We explicitly stated these in Section 2 – SSM and at the beginning of the experiment section. The optimal policy is always validated with Dynamic Programming._
> > > > >
> > > > > _We DID understand the importance and that was why we explicitly stated these in the manuscript._
> > > > >
> > > > > _Considering that the agents have limited learning capacity with a rather small training budget, success rate of 0.4 is in fact very impressive considering the difficulties of the OOD evaluation setting that we have adopted. In the original paper of Skipper, the authors showed that much simpler tasks than SSM can result in very low performance in terms of success rate among the state-of-the-art methods, including Director._
> > > > >
> > > > > `I suggest moving the related work section either after the introduction, or before the conclusion. Now it breaks the flow of the paper since it is placed just before the experiments.`
> > > > >
> > > > > _We have moved the section accordingly. Thank you for your suggestion._
> > > > >
> > > > > ***(TO BE CONTINUED)***

---

> > > > > > ### Author Response · Authors · 2024-11-15
> > > > > > **Reply to Reviewer XvdY (Part 6 / 6)**
> > > > > >
> > > > > > `The authors say in line 371 "Due to page limit, 3 out of 4 sets of experiments are only presented in the Appendix". While I do not expect all of the experiments to fit in the main paper, being able to condense and critically decide what goes in the main text is an important task authors should dedicate time to. This is mainly an empirical paper, and it is thus important for the most relevant experiments to be in the main text such that the claims can be supported by those experiments.`
> > > > > >
> > > > > > _We previously summarized the experiments in the Appendix at the end of the experiment section in the submission._
> > > > > >
> > > > > > _Following your comments, we extended the statements there. Now it reads:_
> > > > > >
> > > > > > _" With the proposed strategies, we saw a reduction in delusional behaviors during training in both Skipper and LEAP (in Appendix), which led to better OOD generalization performance in 2 sets of environments, posing challenges of G.1 and G.2, respectively (more results in Appendix). The 4 sets of experiments align in terms of the conclusions."_
> > > > > >
> > > > > > `'generators are a major source of risk related to delusion' should at least backed up by some previous work.`
> > > > > >
> > > > > > _Thank you for your suggestions, we added two citations regarding the risks of hallucinations._
> > > > > >
> > > > > > `There are also many parts with grammatical errors, e.g. line 186 -line 251 -line 257 etc.`
> > > > > >
> > > > > > _We have made many and large amounts of edits for better communications in the revised manuscript, and tried our best to eliminate all grammatical errors. We hope that you could take some time to see if the problems are fixed to your liking._
> > > > > >
> > > > > > ----------
> > > > > >
> > > > > > **_We are grateful that you have spent huge amounts of time trying to give concrete suggestions to our work and understand that you may have concerns about the claims of this paper and we have tried our best to reply to your comments, with the hope that we could reach a consensus accordingly. We also acknowledge that this paper was not written in a form that you intended it to, but we would like to humbly suggest that you take consideration into our perspectives and_** **_place greater emphasis on the contributions and ideas presented in this work._**
> > > > > >
> > > > > > **_We understand that it could also be the fact that the previously-submitted manuscript did not do a good enough job in terms of communication that caused some confusion. Thus, we spent the past few days and did our best to rewrite it, hopefully to your satisfaction._**
> > > > > >
> > > > > > **_With the utmost sincerity, we have demonstrated that we are grateful for your many suggestions and have invested as much energy as we could to shape the manuscript to your liking. We hope that you could consider raising the review score, since excluding your concerns regarding the scope of our claims, we believe that you fundamentally approve of the contributions we have made._**
> > > > > >
> > > > > > **_We completed this work with the hope that it could help to bring awareness of delusions to our research community so all of us could reduce time and energy wasted on repeating these mistakes (of making delusional agents and not understanding what went wrong), thus accelerate the research and increase the impact of our field._**
> > > > > >
> > > > > > **_Thank you very much for your consideration!_**

---

> ### Comment · Reviewer_XvdY · 2024-11-25
>
> Thanks for the detailed answer and patience.
>
> I suggest the authors highlight changes in their text with a different color. It is very difficult for me to exactly go to all places and find what has been changed, etc. This would make much clearer where my focus should be.
>
> >We chose not to not because we didn’t try to do so, instead, we tried with considerable effort yet could not find a convincing case (a good environment-agent combination) as of now, mostly because target-directed frameworks are under-researched and the appropriate evaluation scenarios are not quite accessible.
>
> Being this a purely empirical paper, I believe it is important to have an extensive empirical evaluation. More environments (which I did not request) do not necessarily mean continuous ones for example, but ones where a particular insight should be gained. The authors should spend time in this.
>
> > even if a generator is trained with autoregressive or unconditional losses, as long as it fits into the target-directed framework, we can apply our strategies onto the estimators to deliver the results.
>
> Can you clarify while the method is applicable, we should all believe that the same results will also hold? A toy experiment where this is the case (i.e. applying your method in a source-target pair scenario which is not HER) would make the claim considerably more credible.
>
> >, we have tried our best to improve on the visual presentations
>
> which images have been changed?
>
> > widely adopted in literature / on goal misgeneralization
>
> please add citations I should look at . I am familiar with other piece of work where your experiments would NOT be OOD. This also stands for other section of your answer, when you mention things coming from other paper but do not cite any. I am familiar with Procgen, but even there I believe the experiments I suggested are sensible. Moreover, I still do not understand why the average task of difficulty is the same and do not try to run it with a wider range
>
> > Following your comments, we extended the statements there. Now it reads:
>
> I suggest the author to include more experiments in the main text, and cutting other parts which may be redundant. Again, the experiments are the main contribution of these paper and in the main text there is only one environment.
>
> > Considering that the agents have limited learning capacity with a rather small training budget, success rate of 0.4 is in fact very impressive considering the difficulties of the OOD evaluation setting that we have adopted.
>
> I disagree. If you argue that your method is that much stronger, I suggest investing the computing budget to showcase runs for a larger training budget ( it should not be 20 seeds obviously). It is hard to verify such a claim otherwise. Moreover, again here you make a strong claim, while in other parts of your answer you say that you are not.
>
> > Your suggested OOD scenario translates to an agent generalizing its learned skills in tasks NOT sharing the same nature with the training tasks, as the agents were never trained to deal with cases where two shields come into play
>
> Good point, but mine was just a suggestion. For example, you can train the agent making it start with already a shield in hand for example. I believe this would be OOD and feasible in your scenario. While your OOD setting is valuable, I believe it is somewhat restricted
>
>
> I tried to address most of your answers. I suggest the authors again in making an answer as digestible as possible - I had to search for the papers you were mentioning, look at the paper again in extreme detail to find the changes, etc. I also appreciate the long answer. However, I suggest having a recap of the main important points you want me to convey or focus; otherwise, it is difficult for me to exactly understand the line of thoughts the authors have in mind.
>
> I sincerely believe this work is very interesting, and has potential. However I do not believe it is in the current shape for being published, mainly given my above concerns. I want to emphasize that I *do no want to ask for excessive experiments*, but that (i) being appropriate in your claims (ii) have proof of concepts experiments, since you are providing a new method and claiming its results, which cannot be taken true otherwise. If this is trivial, it would also be trivial to run some toy experiments to valide this.

---

> ### Author Response · Authors · 2024-11-26
> **Second Round Replies to Reviewer XvdY's New Comments (Part 1 / 4)**
>
> _Thank you very much for your detailed reply._
>
> _Before addressing your concerns, we think it is important to point out that **much of the confusion seems to stem from a misunderstanding of our experimental setup**._
>
> _For each random seed, we generate 50 training environments with a difficulty of 0.4. These environments are fixed throughout the training / evaluation process. At the start of each episode, one of these 50 environments is randomly selected as the training environment. After each episode ends, a new environment is randomly chosen for the next episode._
>
> _When the training hits certain milestones, e.g. every 50,000 steps, we make a clone of the agent and test the clone on 4 sets of newly generated environments, each 20 times, spanning difficulties {0.25, 0.35, 0.45, 0.55}._
>
> _More specifically, this cloned agent is first tested on 20 newly generated, previously unseen environment instances of difficulty 0.25. Then, we collect the datapoints (binary success / fail) and move to difficulty 0.35, then 0.45 and finally 0.55. After collecting all 80 datapoints per seed from totally 20 independent seed runs, we have 1600 points to calculate the mean and the confidence interval at one particular milestone value (x value in Fig.3 - H). To be absolutely clear, each difficulty contributes 400 binary datapoints just for one point on any curve in Fig.3 - H)._
>
> _Importantly, **an agent was only ever trained on the 50 difficulty 0.4 environments and will never see the 80 OOD evaluations environments with difficulty spanning from 0.25 to 0.55. This is why our evaluation procedure is undeniably OOD**._
>
> _Something that seemed to be the source of some confusion was the average task difficulty. The aggregated OOD difficulty matched training, because we have an equal weighting of all difficulties {0.25, 0.35, 0.45, 0.55}. **Note that there was never evaluation performed at difficulty 0.4**. We only matched the **average** task difficulty, so readers could intuitively examine the generalization gap - the performance discrepancy between the training and the evaluation, which is important because it was introduced as one of the motivations of the target-directed approaches._
>
> _Additionally, the aggregated OOD performance was presented **along with the non-aggregated ones for each difficulty** (please check Fig.6 in the Appendix). For each case of OOD difficulty and each x value, these non-aggregated curves have 20 datapoints (binary success / fail) for each random seed. This means all 400 points were collected from 20 random seeds to get their mean and confidence interval to paint just one point (x-value) in the curves of Fig.6 – B)._
>
> _Now that our settings have been clarified, we would like to address your concerns in detail._
>
> ----
>
> `widely adopted in literature / on goal misgeneralization; please add citations I should look at. I am familiar with other piece of work where your experiments would NOT be OOD. This also stands for other section of your answer, when you mention things coming from other paper but do not cite any. I am familiar with Procgen, but even there I believe the experiments I suggested are sensible. Moreover, I still do not understand why the average task of difficulty is the same and do not try to run it with a wider range`
>
> **_On Task Average Difficulty_**_:_
>
> _In this question, you raised concerns about the average task difficulty. We believe this confusion arose from a misalignment in understanding our experimental setup. Hopefully, the clarification above suffices regarding this point._
>
> **_On ProcGen:_**
>
> _We are happy that you are familiar with ProcGen, as it should now be clear that our training-evaluation setting is almost the same as Sec 3.3 in ProcGen (500-levels). The difference is that ours have an emphasis on a wide range of OOD difficulties._
>
> **_Regarding Citations_**_:_
>
> _The statement “widely adopted in literature / on goal misgeneralization” was misquoted. We never claimed that our OOD-focused setting has anything to do with goal-misgeneralization. The text read as follows (in thread Reply to `Reviewer XvdY Part 3/6`):_
>
> > _“The OOD setting we adopted in our work is only a subset of OOD generalization challenges, which we have motivated at the beginning of the paper. Very importantly, we want to emphasize that this setting has been long-accepted and **widely adopted in literature**.”_
>
> _It is possible that parts of our reply regarding the OOD setting were conflated with the parts regarding the connections between our work and the two categories of goal-misgeneralization._
>
> _ProcGen should be a sufficient reference for our OOD setting, given its impact and your familiarity with it._
>
> ***(TO BE CONTINUED)***

---

> ### Author Response · Authors · 2024-11-26
> **Second Round Replies to Reviewer XvdY's New Comments (Part 2 / 4)**
>
> **_On why not run experiments on a wide range_**_:_
>
> _The experiments were indeed conducted on a wide range, from 0.25 to 0.55. We experimentally found that over 0.55, it can be difficult to guarantee the existence of a viable path of success. Lower than 0.25, the task become trivially easy and G.1 targets will disappear._
>
> `Your suggested OOD scenario translates to an agent generalizing its learned skills in tasks NOT sharing the same nature with the training tasks, as the agents were never trained to deal with cases where two shields come into play. Good point, but mine was just a suggestion. For example, you can train the agent making it start with already a shield in hand for example. I believe this would be OOD and feasible in your scenario. While your OOD setting is valuable, I believe it is somewhat restricted`
>
> _We do not understand your concern, as **we have in our work the same experimental setting that you have suggested here**. At the beginning of the experimental section, we said that we will adopt a uniformly random initial state distribution, which will span situations <0, 0>, <1, 0>, <0, 1> and <1,1>. When an agent is initialized in a state in <0, 1>, it is starting with a shield in hand, exactly as you described._
>
> _We hope that there is no more misunderstanding in the experimental procedure, as the related information is present in both the initial manuscript and the revised one._
>
> `Considering that the agents have limited learning capacity with a rather small training budget, success rate of 0.4 is in fact very impressive considering the difficulties of the OOD evaluation setting that we have adopted. I disagree. If you argue that your method is that much stronger, I suggest investing the computing budget to showcase runs for a larger training budget ( it should not be 20 seeds obviously). It is hard to verify such a claim otherwise. Moreover, again here you make a strong claim, while in other parts of your answer you say that you are not.`
>
> _We believe this was a disagreement over a misunderstanding._
>
> _When we talked about small training budget, we meant that 50 training environments is a small budget, not the 1.5M interaction steps, and not about the 20 seeds. We intentionally kept the number of training environments small to increase OOD difficulty._
>
> _Additionally, **when we claimed that the aggregated success rate of ~0.4 is impressive, we made such statements based on the experimental data on hand**._
>
> _With Skipper variant F-(E+P), the average success rate on difficulties 0.25, 0.35, 0.45 and 0.55 are 50%, 42%, 40% and 35%, respectively (Fig. 6). Yet the most well-performed non-target-directed method that we have tested, a Rainbow variant which we optimized for SSM and RDS, produces <10% average success rate in all 4 OOD cases. For RDS, the respective success rates are 92%, 90%, 88%, 81% (Fig. 9), while the best baseline gives <20% consistently in all 4 OOD cases. These are not the only places where we explicitly discussed the difficult nature of these tasks. In experiments with LEAP, we showed that LEAP cannot solve the 12x12 SSM tasks satisfactorily at all, and we had to shrink the environment size to 8x8. We did not include the results of the non-target-directed baselines, since their performance is bad, and they have nothing to do with delusions._
>
> _Our experiments have consistently shown that both Skipper and LEAP do not benefit from longer training than 1.5M steps, as the performance curves show stagnation after 1M steps (Fig.1 – H, Fig. 6 and Fig. 9)._
>
> _As discussed, each part of the aggregated performance curve (with the associating bands) is powered by 400 samples, given by 20 individual seed runs and shows definitive statistical significance in difference. It is unreasonable to think that this is not a strong claim._
>
> `even if a generator is trained with autoregressive or unconditional losses, as long as it fits into the target-directed framework, we can apply our strategies onto the estimators to deliver the results. Can you clarify while the method is applicable, we should all believe that the same results will also hold? A toy experiment where this is the case (i.e. applying your method in a source-target pair scenario which is not HER) would make the claim considerably more credible.`
>
> _Our disagreement here may stem from a lack of clarity regarding the interchangeability of HER methods with other source-target pair-based approaches. For this point, we added more explanations in the revised manuscript._
>
>
> ***(TO BE CONTINUED)***

---

> ### Author Response · Authors · 2024-11-26
> **Second Round Replies to Reviewer XvdY's New Comments (Part 3 / 4)**
>
> _Source-target pairs can be assembled beforehand, like in HER, or be assembled just-in-time, by sampling two transitions from any ordinary ER. There is NO DIFFERENCE between training procedures with or without HER, as long as the training is based on source-target pairs._
>
> _This immediately leads to two scenarios:_
>
> _1) We can apply our HER based strategies on any non-HER method trained on source-target pairs. This can be done by introducing an additional HER._
>
> _2) We can apply our strategies in a way equivalent to scenario 1) without the introduction of HER to generic non-HER method trained on source-target pairs._
>
> _Once these points are understood, the suggested toy-experiment would not be necessary._
>
> _We never intended to say that our method would produce the same results on other methods, as you are suggesting. We said instead “to deliver the results”. Different target-directed methods are impacted by delusions differently. Therefore, addressing delusions in these methods will have different results, some significant, some negligible. We were very aware of this fact and emphasized that our method is **APPLICABLE** to generic target-directed methods._
>
> `“We chose not to not because we didn’t try to do so, instead, we tried with considerable effort yet could not find a convincing case (a good environment-agent combination) as of now, mostly because target-directed frameworks are under-researched and the appropriate evaluation scenarios are not quite accessible.” Being this a purely empirical paper, I believe it is important to have an extensive empirical evaluation. More environments (which I did not request) do not necessarily mean continuous ones for example, but ones where a particular insight should be gained. The authors should spend time in this.`
>
> _You suggested that our experiments are not extensive. On this point, with all respect, **we sincerely feel that this comment is not justified and not appropriate given our endeavors**._
>
> _We have tried our best to acquire insights through our experiments, and that was why we have crafted the two environments and made sure that all aspects of the experiments can be validated through ground truth solved by DP. Respectfully, almost no other existing literature backs up their hypotheses about behaviors of their agent by analytical metrics like this work. We did them precisely because we were careful about the experimental rigor. Reviewer 2JsJ explicitly commended us for this._
>
> _We also think part of your concern is misaligned with our previous replies to you. Our response regarding LEAP was not about the environment (continuous or not), rather the **limited number of target-directed methods** that can be used to gain more insights. We spent months adapting two methods convincingly. The reason why we asked ourselves to have two methods, was to show that delusions impact different target-directed methods in different ways. Some methods are more affected by delusions, for example, delusions damage LEAP significantly more than Skipper, as shown in experiments._
>
> _While we remain truly open to constructive suggestions, we would greatly appreciate it if you could give some concrete explanations on why you think our experiments were not extensive nor insightful enough._
>
> `Following your comments, we extended the statements there. Now it reads ... I suggest the author to include more experiments in the main text, and cutting other parts which may be redundant. Again, the experiments are the main contribution of these paper and in the main text there is only one environment.`
>
> *In the revision, we included extended discussions about the experiments and shrunk many parts of the earlier sections.*
>
> *In the current shape of the paper, it is not possible to include another set of experiments in the main parts of the manuscript without passing the page limit.*
>
> *Despite the experiments playing an important role in this work, the most important sections to us were the introductions to the perspective of delusions, the categorization and the mitigation strategies. This work in no way fits the stereotype of coming up with some trivial experimental techniques and testing them extensively.*
>
> `We have tried our best to improve on the visual presentations ...  which images have been changed?`
>
> _You suggested that Fig. 3 was too condensed, so we changed the layouts of the legends to make them smaller and have less overlap with the figures. We applied the same changes to all performance figures in the revised manuscript._
>
> ***(TO BE CONTINUED)***

---

> > ### Author Response · Authors · 2024-11-26
> > **Second Round Replies to Reviewer XvdY's New Comments (Part 4 / 4)**
> >
> > `I suggest the authors highlight changes in their text with a different color. It is very difficult for me to exactly go to all places and find what has been changed, etc. This would make much clearer where my focus should be.`
> >
> > *We are sorry about the inconvenience. OpenReview has notified us that a `pdfdiff` will be applied automatically to highlight the differences during the revision. Could you check if a `pdfdiff` is automatically generated at the end of the new revision?*
> >
> > *We unfortunately cannot change the text color since ICLR requires all manuscripts to go strictly according to the publishing guidelines. We will provide the following summaries of the changes during revision:*
> >
> > _We heavily edited the manuscript, writing for better motivated story (while making cuts mostly for more contents in the experiment section):_
> >
> > _Target-directed agents like chasing problematic targets -> they are acting delusional -> delusional behaviors come from delusions -> understand the nature of target-directed methods trained on source-target pairs -> how do delusions form and where do they reside -> how to deal with them -> experiments -> related works section: discuss why people rarely have taken this perspective -> guidelines for designing non-delusional agents -> conclusion_
> >
> > _Particularly for the experiment section, we added more explanations regarding the setup and the intentions of certain experimental procedures and statements, following the suggestions of you and Reviewer 2JsJ._
> >
> > ----
> >
> > _**As always, we are grateful for your comments and sincerely hope this reply will address your concerns sufficiently, so that you may reconsider your rating of this work.**_

---

> > > ### Comment · Reviewer_XvdY · 2024-12-03
> > >
> > > Thanks for your detailed answer
> > >
> > > > We unfortunately cannot change the text color since ICLR requires all manuscripts to go strictly according to the publishing guidelines. We will provide the following summaries of the changes during revision:
> > >
> > > pdfdiff does not work for me (I cannot compare with the first version), and is also sometimes unreliable. I would like to say that it is common for authors to highlight the changes in the rebuttal period, to help reviewers who are trying to understand better your changes and care about giving a high-quality review. You would then have changed it to the right color for the camera-ready version.
> > >
> > > > You suggested that Fig. 3 was too condensed, so we changed the layouts of the legends to make them smaller and have less overlap with the figures. We applied the same changes to all performance figures in the revised manuscript.
> > >
> > > I still find the images very confusing. Plots should be completely SELF contained, but this is not the case in your paper. I think the presentation of the result can much improved, as also pointed out by other reviewers.
> > >
> > > > We were very aware of this fact and emphasized that our method is APPLICABLE to generic target-directed methods.
> > >
> > > I agree with this, but my impression of your claims was that you were claiming that your method was also gonna work for other training methods ( and if you do not have experiments to show that, this cannot be claimed). E.g., this is the impression I got from this answer " we can apply our strategies onto the estimators to deliver the results.".
> > >
> > > > Note that there was never evaluation performed at difficulty 0.4...
> > >
> > > I see this, and my point is partially clarified. However, in the breakdown runs, I still see a big drop in performance even when going to the easiest method (0.25), suggesting that there is room for improvement. Clearly you beat the baseline and I am not denying that, but this cannot be the only metric to judge a work! But I see better your claim, thanks for clarifying
> > >
> > > > we have in our work the same experimental setting that you have suggested here...
> > >
> > > In the paper, I read this "To maximize difficulty, the initial state is
> > > fixed in each novel evaluation task instance: the agents are not only spawned to be at the furthest
> > > side of the monster, but also in semantic class ⟨0, 0⟩, " which contradicts what you say there. Can you explain this?
> > >  Let me clarify again: train with agent spawning in state <0,0> always, then test with agent spawning in <1,0> and report the performance. I believe a result like that would be interesting to have, and I do not understand the aversion of the authors towards my suggestion.
> > >
> > >
> > >
> > > I will raise my score to a 5, recognizing the effort and the potential of the work. I believe that the presentation and the clarity of the result can be vastly improved. I warmly suggest the authors to be more receptive of reviewers' feedback in the future.

---

> > > > ### Author Response · Authors · 2024-12-04
> > > > **Third Round Replies to Reviewer XvdY's New Comments (Part 1 / 2)**
> > > >
> > > > ***We are happy to have addressed most of your raised concerns and that you have raised the score. Please see our replies to your latest raised concerns.***
> > > >
> > > > ---
> > > >
> > > > `pdfdiff does not work for me (I cannot compare with the first version), and is also sometimes unreliable. I would like to say that it is common for authors to highlight the changes in the rebuttal period, to help reviewers who are trying to understand better your changes and care about giving a high-quality review. You would then have changed it to the right color for the camera-ready version.`
> > > >
> > > > _Thank you very much for sharing this with us._
> > > >
> > > > _We were not aware highlighting was an option before, but we have now marked the important changes in the revision with purple fonts._
> > > >
> > > > _Since manuscript revision is currently disabled, we share it with you with the following anonymized link:_
> > > >
> > > > *https://github.com/AnonymousAuthors21/ICLR2025-rebuttal/blob/main/delusions_revision_colored2.pdf*
> > > >
> > > > _We marked the changes by hand after rendering the PDF, so there could be slight distortions to the font. This won’t be an issue in future versions._
> > > >
> > > > `I still find the images very confusing. Plots should be completely SELF contained, but this is not the case in your paper. I think the presentation of the result can much improved, as also pointed out by other reviewers.`
> > > >
> > > > _We understand that normally figures are self-contained. However, the page limit and the fact that we need many figures to be convincing required that we distribute the legends into subfigures. This was a compromise we did not want to make but that was needed in the end. We are actively seeking better ways to organize the figures, and wonder if you might have any suggestions?_
> > > >
> > > > `Note that there was never evaluation performed at difficulty 0.4... I see this, and my point is partially clarified. However, in the breakdown runs, I still see a big drop in performance even when going to the easiest method (0.25), suggesting that there is room for improvement. Clearly you beat the baseline and I am not denying that, but this cannot be the only metric to judge a work! But I see better your claim, thanks for clarifying`
> > > >
> > > > *We understand your concern. However, please allow us to point to the following facts:*
> > > >
> > > > ***[We seek statistically significant improvement with multiple metrics]**: Our work is focused on mitigation strategies that _enhance_ the performance of baseline agents by addressing their delusional behaviors. **Thus, the significance of this work should be determined by if the mitigation strategies could bring statistically significant improvements to the performance of the baselines**; Contrary to your claim, we used several metrics to judge the improvement. For example, we measured the delusional errors and the frequencies of delusional behaviors for both baseline methods. Please kindly check the manuscript to verify this.*
> > > >
> > > > ***[OOD Evaluation was Deliberately Made Challenging]**: In our main experiment, we used Skipper on the 12x12 SSM, where you observed a sharp drop in performance. Please note that **this drop is intentional and not a limitation**. As we discussed in the introduction, target-directed methods are specifically designed to address the "generalization gap" between training and evaluation, a gap that both we and you have noted. However, these gaps are not always significant in our work. For example, in Figure 9, which shows Skipper on the 12x12 RDS, the gap is much smaller.*
> > > > *The generalization gap is influenced by both the agent and the environment. When you mentioned that "there is still room for improvement," it’s important to consider that in 2 out of 4 experimental settings, the baselines already exhibited small generalization gaps. To highlight the gap, we intentionally used a large world size (12x12) in the SSM. We understand that you would like a baseline method that performs well on SSM as well; however, our goal was to intentionally challenge the agents. If we reduced the world size, the gap would naturally shrink.*
> > > >
> > > > ***(TO BE CONTINUED)***

---

> > > > > ### Author Response · Authors · 2024-12-04
> > > > > **Third Round Replies to Reviewer XvdY's New Comments (Part 2 / 2)**
> > > > >
> > > > > `In the paper, I read this "To maximize difficulty, the initial state is fixed in each novel evaluation task instance: the agents are not only spawned to be at the furthest side of the monster, but also in semantic class ⟨0, 0⟩," which contradicts what you say there. Can you explain this? Let me clarify again: train with agent spawning in state <0,0> always, then test with agent spawning in <1,0> and report the performance. I believe a result like that would be interesting to have, and I do not understand the aversion of the authors towards my suggestion.`
> > > > >
> > > > > *Let us explain why there is no contradiction and how our initializations differ from your clarification.*
> > > > >
> > > > > *We TRAIN the agents by initializing them in all 4 situations: <0, 0>, <0, 1>, <1, 0>, <1, 1>. (Section 5.1, Paragraph 2)*
> > > > >
> > > > > *We EVALUATE the agents by initializing only in <0, 0> (with a range of OOD difficulties). (Section 5.2, Improvements on OOD Performance)*
> > > > >
> > > > > *As you can see, our train/evaluate setting is the REVERSE of what you suggested.*
> > > > >
> > > > > *Here is why we think our setting is more appropriate.*
> > > > >
> > > > > *1. By uniformly initializing according to situations, we allow the agents to have adequate exploration, and to accelerate training (shorten the number of agent-environment interactions needed for experiments and save time). This is not only because that exploration is not the focus of this work, but also because our preliminary experiments showed that using the <0, 0>-only initialization does not affect the final performance significantly (only making the convergence longer, will need 0.5M steps more during training), therefore wasting significantly more resources.*
> > > > >
> > > > > *2. For an agent to succeed in the evaluation tasks, it must take one of the two paths: <0, 0> to <1, 0> to <1, 1> (gather sword first) or <0, 0> to <0, 1> to <1, 1> (gather shield first). Since the tasks are all Markovian, after the agent successfully transitions from situation <0, 0> to situation <1, 0>, the rest of the task is _equivalent_ to initializing a task in <1, 0>. In other words, initializing a task in <1, 0> is a subtask of those initializing in <0, 0> and your suggested cases are being tested.*
> > > > >
> > > > > ***We hope this demonstrates that we understand your clarified suggestion and we certainly do not have any aversion to your propositions.***
> > > > >
> > > > > -----
> > > > >
> > > > > **_Again, we are grateful for your comments and sincerely hope this reply sufficiently addresses your concerns._**

---

### Official Review · Reviewer_2JsJ · 2024-11-03

**Soundness:** 3
**Presentation:** 2
**Contribution:** 3
**Rating:** 5
**Confidence:** 3

**Summary:**

This paper studies the problem of “delusions” in target-directed (goal-conditioned) RL by establishing a taxonomy of delusions that are possible for the Generator (proposer of the targets) and the Estimator (that measures the goodness of the target). Then, they propose two new HER relabeling strategies (and mixed strategies) to mitigate the effects of different types of delusions.
Finally, they evaluate their proposed strategies in a controlled gridworld that allows them to quantify the different kinds of delusional targets they consider.

**Strengths:**

1. The systematic approach to studying delusions in target-oriented RL is relevant to the community. The authors' approach allows us to understand the potentially problematic behavior that can arise from naively training agents within this paradigm.
2. The empirical evaluation shows an organized approach to verifying the effect of the types of delusions and the effects of potential mitigation strategies.
3. The knowledge provided in the paper is of general interest to practitioners and researchers.

**Weaknesses:**

1. I can see that the authors studied the problem with empirical rigor, however, the experimental section, which goes a long way to showcase the problems, looks a bit crammed and it’s hard to follow. I would suggest sacrificing part of the previous discussion to treat the empirical section with more care.
2. Figure 3, it’s hard to follow because of the number of overlapping curves, colors, and legend positioning. This overall makes the curves hard to interpret.
3. Though I understand that scaling these ideas to more complex environments can be the subject of future research, perhaps showcasing the effect on the learning performance these mitigation strategies can have in a different environment (beyond the controlled gridworld proposed) would also be informative and relevant to this paper.

**Questions:**

1.  I see much of the mitigation strategies are based on making the estimator better and recognizing delusions from the generator. What kind of mitigation strategies (beyond "generate") would be useful to generate less delusional targets?

---

> ### Author Response · Authors · 2024-11-15
> **Reply to Reviewer 2JsJ (Part 1 / 2)**
>
> **_Thank you very much for the time and energy you have spent reviewing this paper._**
>
> **_To reply to you with deserving respect, we have organized your comments point-by-point so we will not leave ANY of your points behind and try our best to address all your concerns._**
>
> ----------
>
>  `I can see that the authors studied the problem with empirical rigor, however, the experimental section, which goes a long way to showcase the problems, looks a bit crammed and it’s hard to follow. I would suggest sacrificing part of the previous discussion to treat the empirical section with more care.`
>
> _We have followed your suggestions to cut down on the discussions before the experiments and add more explanations to the experimental section. Specifically, we added intuitive guiding sentences to make the reader understand our progression in the experimental analyses. We also incorporated the suggestions from the other reviewers in this section. Please consider reading the updated manuscript. Specifically, we made sure that the intention of each result analysis was clearly stated._
>
> _Thank you very much for your suggestions, we sincerely appreciate them._
>
> `Figure 3, it’s hard to follow because of the number of overlapping curves, colors, and legend positioning. This overall makes the curves hard to interpret.`
>
> _We understand your concern. The visual difficulty was caused by the fact that we have quite a relatively large number of baselines and variants (that we wanted to discuss at the same time)._
>
> _Following your suggestions, we tried to optimize the layouts of the figures and make the visuals clearer. Specifically, we tried to shrink and distribute the legends to subfigures that minimally cover the curves and legends. We applied these changes to figures across the manuscript, including the Appendix._
>
>
> `Though I understand that scaling these ideas to more complex environments can be the subject of future research, perhaps showcasing the effect on the learning performance these mitigation strategies can have in a different environment (beyond the controlled gridworld proposed) would also be informative and relevant to this paper.`
>
> _We appreciate your understanding of the contributions of this paper and value your suggestions. We wish to communicate with you the reason why we didn’t incorporate the additional results you are suggesting here: we tried, but they were not appropriate enough._
>
> _We put significant effort into trying to showcase learning performance improvement in more complex environments. However, since target-directed frameworks are still very under-investigated in RL, very few existing methods can be used to begin with. The most challenging aspect about this is that these existing methods seem too hyperparameter- or architecture-sensitive to be deployed to other environments (than the ones these agents are fine-tuned to), where delusions could be consequential enough to cause problems._
>
> _Let us provide you with an example of our endeavors on trying out our ideas on LEAP with the mujoco-based continuous-action navigation task. The original LEAP agent was implemented on a single task setting where OOD evaluation is neglected. However, after we implemented our mitigation strategies for LEAP (trivially easy since our mitigation strategies are simplistic) and tried to change the mujoco environments s.t. the walls become deadly to agents (increase the consequences of delusions), the end results did not make us feel comfortable to include them, for the following reasons:_
>
> _**[Not Convincing]** The results do not link to OOD generalization in any way and the environment cannot be diagnosed with ground truth access. We will not be able to prove if any performance improvement is indeed caused by decreasing delusions;_
>
> _**[Inappropriate Environments]** The Ant-Navigation environment, which LEAP was fine-tuned to, has a fully reversible state space. This means G.2 targets will not exist and the G.1 targets will not result in termination of episodes or any punishment, making the final performance, the only metric that is accessible to us, not sensitive to delusions._
>
> ***(TO BE CONTINUED)***

---

> > ### Author Response · Authors · 2024-11-15
> > **Reply to Reviewer 2JsJ (Part 2 / 2)**
> >
> > _**[Existing Methods Are Too Sensitive]** We found that even making minimal changes to the training environment completely breaks the agent’s baseline performance. We wanted to make the environment more dangerous for G.1 targets, such that the episode gets terminated when the agent pushes itself into a wall. However, after months of trying, we were not able to get the baseline performance to anything more significant; To include G.2, we also tried to port the baseline to Ant-Push, where a previous push of block permanently changes the environment. This adaptation was also met with failure, as the baseline would not work. We believe the difficulties in adaptation are generally due to the existing target-directed agents being over-finetuned. It was only after this failure, we started to transplant LEAP into our own environment SSM, where we could isolate the additional irrelevant difficulties and, in the end, succeeded in producing a baseline with convincing performance in a more convincing setting._
> >
> > _As authors working in an academic setting with limited computational resources, we were unable to generate additional compelling results in time for the manuscript submission. We are continuing to work on this aspect and appreciate your understanding of the challenges involved in meeting such resource-intensive demands._
> >
> > _At a high-level, the difficulty of presenting more experimental results was caused by the chicken-and-egg problem, which refers to the fact that target-directed agents are still novel to the field and are under-researched while problems like delusions hinder the research into this direction. We believe that the contribution of this work can be used to break the performance curse of target-directed agents, s.t. better performance will incentivize more research interests in this direction and let us break away from this chicken-and-egg situation._
> >
> > `I see much of the mitigation strategies are based on making the estimator better and recognizing delusions from the generator. What kind of mitigation strategies (beyond "generate") would be useful to generate less delusional targets?`
> >
> > _Thank you for your insightful questions._
> >
> > _The question you asked here was “what mitigation strategies would be useful to generate less delusional targets”._ _We believe that, since the manuscript discusses mitigation strategies aimed solely at reducing estimator delusions, your question may be asking how to effectively address hallucination during generation. We hope this interpretation is correct; if not, please feel free to clarify during the discussion period._
> >
> > _First, evidently, an appropriate inductive bias that suits the observation / state space of the environment in the generator architecture would be significant in reducing G.1 targets. A good inductive bias focuses the generation onto spaces that are more aligned with the semantic rules of the environment. For example, if the targets are meant to be colored image observations, using a greyscale generator would make most targets G.1._
> >
> > _Second, as discussed in the manuscript, because of the existence of G.2 targets, training procedures need to be taken with care. This means, unless useful, we should not let the generator to be trained on <state, G.2 target> pairs, such that the generator would not waste its learning capacity to learn how to generate G.2 targets._
> >
> > _Finally, because of the uncontrollable nature of generalization in neural networks and other inductive biases capable of generalization beyond tabular lookups, hallucination is generally unavoidable._
> >
> > _Following your question, we have improved the writing in the manuscript in the corresponding locations such that readers would get a clearer understanding of this problem upon reading. Thank you for your valuable comment!_
> >
> > ----------
> >
> > **_We have done our best to address all your concerns thoroughly and, we hope, to your satisfaction. Our sincere intent in completing this research is to bring greater awareness of delusions in agent behavior to the research community. We hope this work will help reduce the time and energy spent on recurring issues, such as creating delusional agents and misunderstanding their failures, thereby accelerating research progress and enhancing the impact of our field._**
> >
> > **_We sincerely wish that you could consider raising the review score, since we believe that you approve the contributions of this work, and the score is barely below the targeted acceptance threshold._**
> >
> > **_Thank you very much for your consideration!_**

---

> > > ### Comment · Reviewer_2JsJ · 2024-11-27
> > >
> > > Thank you for taking the time to provide detailed answers!
> > > Though I appreciate the efforts made by the authors, I still believe the paper would benefit from an improved presentation of the results. For instance, considering different types of visualizations, because I believe that some of the plots (e.g., Fig 3g/3c) do not provide a lot of information in the form of learning curves.
> > > I will be keeping my score, as I believe that the content of the paper is interesting but it would have a better impact if the presentation was improved.

---

> > > > ### Author Response · Authors · 2024-11-27
> > > > **Second Round Reply to Reviewer 2JsJ**
> > > >
> > > > Thank you for your detailed feedback. We would appreciate clarification on a few points:
> > > >
> > > > Could you elaborate on why you feel Sections 3c and 3g are not informative?
> > > >
> > > > *We’d like to remind you that additional curves and analyses are included in the appendix, which might address your concerns.*
> > > >
> > > > Additionally, if the presentation is a key concern hindering acceptance, we would be grateful if you could suggest specific steps or improvements to better address this issue.
> > > >
> > > > *We have made significant efforts to incorporate your feedback and adapt our experiments accordingly, and your guidance here would be invaluable.*
> > > >
> > > > ***Thank you for your time and input!***

---

### Meta-Review · Area_Chair_NuVB · 2024-12-21

**Metareview:**

The paper addresses the problem of delusions in target-directed RL agents, where agents develop false beliefs about their targets. The authors categorize delusions into target-related and estimator-related errors. Then, the authors proceeded to propose two new strategies—'generate' and 'pertask'—for mitigating delusions, as well as a hybrid approach for separating generator and estimator training. The paper is empirically validated in controlled environments (SSM and RDS) using Hindsight Experience Replay (HER) and includes ablation studies and performance metrics.

Reasons to accept
- The paper tackles a unique challenge in RL, addressing delusional behaviors in target-directed decision-making, which has not been explicitly explored in prior work to the best of my knowledge.
- The categorization of delusions is well-structured, providing actionable insights into RL challenges.
- The paper presents thorough empirical validation with clear visualizations, and the hybrid approach to separating generator and estimator training demonstrates a good understanding of the problem.

Reasons to reject
- The experiments are confined to controlled, grid-world environments (SSM and RDS). It is unclear if the proposed method can be applied to more complex or real-world scenarios.
- The paper lacks a formal theoretical analysis or guarantees about the effectiveness of the proposed strategies, making it difficult to understand why they work and their potential scalability.
- The proposed strategies, especially 'generate,' require significant computational resources, which may limit their applicability in real-world environments with continuous action spaces.
- The paper is conceptually dense and contains unclear or complex terminology, which may make it difficult for readers to follow, particularly those not deeply familiar with RL and HER.

Despite its novelty and contributions, the reviewers are unanimously concerned with the generalization and scalability of this work. Also, the lack of formal analysis, limited experimental scope, and unclear experimental results make it difficult to confidently assess the effectiveness of the proposed method beyond controlled settings. Consequently, I recommend rejecting the paper.

**Additional Comments On Reviewer Discussion:**

During the rebuttal period, two reviewers acknowledged the author's rebuttal, and one reviewer adjusted the score accordingly.

---

### Decision · Program_Chairs · 2025-01-22

Reject